# An Optimal Elimination Algorithm for Learning a Best Arm

**Avinatan Hassidim**
Bar-Ilan University and Google
avinatan@cs.biu.ac.il

**Ron Kupfer**
The Hebrew University of Jerudalem
ron.kupfer@mail.huji.ac.il

**Yaron Singer**
Harvard University
yaron@seas.harvard.edu

## Abstract

We consider the classic problem of $(\epsilon, \delta)$-PAC learning a best arm where the goal is to identify with confidence $1 - \delta$ an arm whose mean is an $\epsilon$-approximation to that of the highest mean arm in a multi-armed bandit setting. This problem is one of the most fundamental problems in statistics and learning theory, yet somewhat surprisingly its worst case sample complexity is not well understood. In this paper we propose a new approach for $(\epsilon, \delta)$-PAC learning a best arm. This approach leads to an algorithm whose sample complexity converges to *exactly* the optimal sample complexity of $(\epsilon, \delta)$-learning the mean of $n$ arms separately and we complement this result with a conditional matching lower bound. More specifically:

- The algorithm's sample complexity converges to *exactly* $\frac{n}{2\epsilon^2} \log \frac{1}{\delta}$ as $n$ grows and $\delta \geq \frac{1}{n}$;
- We prove that no elimination algorithm obtains sample complexity arbitrarily lower than $\frac{n}{2\epsilon^2} \log \frac{1}{\delta}$. Elimination algorithms is a broad class of $(\epsilon, \delta)$-PAC best arm learning algorithms that includes many algorithms in the literature.

When $n$ is independent of $\delta$ our approach yields an algorithm whose sample complexity converges to $\frac{2n}{\epsilon^2} \log \frac{1}{\delta}$ as $n$ grows. In comparison with the best known algorithm for this problem our approach improves the sample complexity by a factor of over 1500 and over 6000 when $\delta \geq \frac{1}{n}$.

## 1 INTRODUCTION

In this paper we study the classic problem of $(\epsilon, \delta) - $ PAC learning a best arm. In this problem there is a set $A$ of $n$ arms and sampling an arm $a \in A$ generates a random variable $\xi(a)$ drawn from some unknown distribution $\mathcal{D}(a) \subseteq [0, 1]$[1]. The mean of every arm $a$ is denoted $\mu(a)$ and an *optimal* arm is $a^\star \in \arg\max_{a \in A} \mu(a)$. A strategy $(\epsilon, \delta)$-learns the best arm if it returns $a \in A$ s.t. $\mu(a) \geq \mu(a^\star) - \epsilon$ with confidence at least $1 - \delta$ over the arm distribution and randomization of the strategy. The goal is to $(\epsilon, \delta)$-learn the best arm with minimal worst case sample complexity over all distributions in $[0, 1]$.

By the celebrated Hoeffding bound we know that it suffices to sample each arm $\frac{1}{2\epsilon^2} \log \frac{1}{\delta}$ times to ensure we are $\epsilon$-close to its true mean with confidence $1 - \delta$, and that without additional information this bound is optimal. A trivial solution is then to estimate the mean of each arm using sufficiently-many samples and take the arm whose empirical mean is largest. A trivial upper bound for learning a best arm using this approach is $\frac{2n}{\epsilon^2} \log \frac{n}{\delta}$.

In a seminal paper, Even-Dar et al. considered the problem of $(\epsilon, \delta)$-learning a best arm when the number of arms $n$ is asymptotically large [12]. They introduce MEDIAN ELIMINATION which is an $(\epsilon, \delta)$-learning strategy whose sample complexity is $\mathcal{O}\left(\frac{n}{\epsilon^2}\log\frac{1}{\delta}\right)$. To date, MEDIAN ELIMINATION is the best algorithm for provably $(\epsilon, \delta)$-learning a best arm in terms of sample complexity when $n$ is sufficiently large. As such it is a fundamental building block in a variety of algorithms (see e.g. [20, 22, 32, 18, 7]), and has applications in a broad range of domains. Unfortunately, the constant terms hiding in the $\mathcal{O}$ notation of the sample complexity of MEDIAN ELIMINATION are quite large. For $n = 100$ its sample complexity exceeds $1000 \times (\frac{n}{\epsilon^2}\log\frac{1}{\delta})$, and grows to over 3 times as $n$ grows.

In terms of lower bounds, the best known bound for this problem is by Manor and Tsitisklis who show that $\frac{n}{128\epsilon^2}\log\frac{1}{4\delta}$ samples are necessary for $(\epsilon, \delta)$-learning a best arm [27]. Thus, the gap between the best known upper and lower bounds exceeds 300,000 and begs the obvious question:

*What is the optimal sample complexity of PAC learning a best arm?*

**Main contributions.**    In this paper we address this question and take fundamentally new approaches to obtain upper and lower bounds for $(\epsilon, \delta)$-learning a best arm. At a high level, our algorithms are designed so that their probability of failure diminishes as the number of arms grows. For a lower bound, we observe that our algorithm as well as many other algorithms for learning a best arm in the literature can be broadly characterized as iteratively sampling and discarding arms until one arm is left. We call algorithms that fit this description *elimination algorithms* and prove a tight lower bound on this class that matches our upper bound. Our results can be summarized as follows:

1. We describe a new algorithm whose sample complexity converges with $n$ to *exactly* $\frac{n}{2\epsilon^2}\log\frac{1}{\delta}$ when $n \geq \frac{1}{\delta}$. This bound *exactly* matches the sample complexity of $(\epsilon, \delta)$-PAC learning the mean of each arm separately according the Hoeffding bound. In comparison to MEDIAN ELIMINATION the sample complexity is lower by a factor greater than 6000 when $n$ is large. Namely, for any given $\lambda < 1$ there is a $\delta_0$ such that for any $\delta < \delta_0$ and $n > 1/\delta$ there exist an algorithm that $(\epsilon, \delta)$-learns a best arm with sample complexity :

$$\left(1 + \lambda\right)\frac{n}{2\epsilon^2}\log\frac{1}{\delta}.$$

2. When $n$ is independent of $\delta$, we describe a simplified version of the algorithm whose sample complexity converges to $\frac{2n}{\epsilon^2}\log\frac{1}{\delta}$; Namely, for any $\lambda > 0$ there exist $\delta_0$ and $n_0$ such that for any $\delta < \delta_0$ and $n \geq n_0$, there exist an algorithm that $(\epsilon, \delta)$-learns a best arm with sample complexity at most: $\left(2 + \lambda\right)\frac{n}{\epsilon^2}\log\frac{1}{\delta}$. Furthermore, for any $\delta < 0.05$, any $n > 0$ and $\epsilon \in (0, 1)$ our approach yields an algorithm whose sample complexity is $\frac{18n}{\epsilon^2}\log\frac{1}{\delta}$. In comparison to MEDIAN ELIMINATION this reduces the sample complexity by a factor greater than 300.

3. Lastly, we prove a matching lower bound; For every $\beta > 0$ there exist $\epsilon_0, \delta_0$ such that for any elimination algorithm which finds an $\epsilon$ best arm with success probability $1 - \delta$ where $\epsilon < \epsilon_0$, $\delta < \delta_0$, there exist $n_0$ such that if $n > n_0$, the algorithm requires at least $\left(\frac{1}{2} - \beta\right)\frac{n}{\epsilon^2}\log\frac{1}{\delta}$ queries.

Technical overview and central insights that led to these results are presented in Appendix G. Our results are in the standard $(\epsilon, \delta)$-PAC learning model, i.e. the goal is to find an $\epsilon$-best arm with probability $1 - \delta$ and sample complexity is measured in the worst case across any distribution in $[0, 1]$ (or any subgaussian, see Appendix E). Before moving forward, it would be instructive to discuss this problem setting as well as closely related settings. Additional related work is in Appendix H.

**Learning an $\epsilon$-best arm.**    As the state-of-the-art algorithm for $(\epsilon, \delta)$-PAC learning a best arm, ME-DIAN ELIMINATION is widely used as a sub-procedure (e.g. [20, 22, 32, 18, 7, 30]). An improvement on its sample complexity as suggested here achieves dramatically lower sample complexity for all procedures that employ MEDIAN ELIMINATION. The interesting regime in this problem setting is the one where $n$ is large, as otherwise it suffices to use the naive sampling strategy of sampling each arm with approximation $\frac{\epsilon}{2}$ and confidence $\frac{\delta}{n}$ and selecting the arm with largest empirical mean.[2]

**Learning an exact best arm.**   In the exact best arm learning problem the goal is to $(0, \delta)$-PAC learn the best arm (see e.g. [3, 22, 18, 17, 29, 28, 14]). This problem is computationally more demanding as arm means can be arbitrarily close and one seeks optimal sample complexity that depends on the arm distributions. For exact best arm learning several algorithms use $\epsilon$-best arm learning as a subroutine, where our work is directly applicable (e.g. [22, 18, 17]). For exact best arm learning, the optimal sample complexity bounds for exponential distributions is achieved in [14].

**Instance based analysis.**   The nature of exact best arm learning necessitates specific assumptions about the relevant families of distributions for the arms. This motivates a series of works that deviate from the $(\epsilon, \delta)$-PAC learning setting where the sample complexity is worst case across all distributions. In particular, a recent line of work analyzes the sample complexity as a function of the given instance (i.e. set of distributions) and $\delta$ for both exact best arm and $\epsilon$-best arm problems [14, 10, 15, 9]. In this genre, variants of *explore and exploit* algorithms known as *track-and-stop* algorithms turned out to be efficient in the number of samples under some assumptions. For for $\epsilon$-best arm, an instance-based optimal algorithm was shown in [15] under the assumption that there is such a unique arm. Recently, [9] show how to generalize this approach without assuming a unique $\epsilon$-best arm. By using a function $T(\bar{\mu})$ from set of distributions to the reals, they show that for any instance $\bar{\mu}$ which belongs to the one-parameter one-dimensional canonical exponential family, $(1 + o(1))T(\bar{\mu})\log\frac{1}{\delta}$ samples are necessary and sufficient for $(\epsilon, \delta)$-learning a best arm, when $n$ is fixed and $\delta$ goes to $0$.

**From instance-based to worst case analysis.**   When the number of arms $n$ is fixed and $\delta$ goes to $0$ and the distribution is bounded in $[0, 1]$, a worst case sample complexity bound can be trivially achieved via the naive elimination strategy. Thus, while this is an interesting regime for instance-based analysis, it is not interesting for worst case analysis. On the other hand, when fixing $\delta$ and letting the number of arms grow, it is not clear what is the asymptotic sample complexity of the problem in worst case, and it cannot be deduced from the instance based analysis. The main contribution of our work is showing upper and lower bounds for this problem.

**Running time.**   Beyond worst case vs. instance based guarantees, elimination algorithms are exponentially faster compared to other approaches like track-and-stop. The algorithms we present here run in $\mathcal{O}(\log^2 n)$ parallel time in the PRAM model [16], hence giving a total implementation in poly-logarithmic time complexity which is an exponential improvement compared to [15, 9].

## 1.1   Paper organization

We present our algorithms in order of increasing complexity. The first is the SIMPLE APPROXI-MATE BEST ARM algorithm introduced in Section 2 which makes assumptions about the input. In Section 3 we present APPROXIMATE BEST ARM which removes these assumptions and achieves sample complexity $\frac{18n}{\epsilon^2}\log\frac{1}{\delta}$ for $\delta < 0.05$ and any $n$ which easily generalizes to achieve a bound that converges to $\frac{2n}{\epsilon^2}\log\frac{1}{\delta}$ as $n$ grows. In Section 4 we present the APPROXIMATE BEST ARM LIKELIHOOD ESTIMATION BY HOEFFDING whose sample complexity asymptotically matches the Hoeffding bound of estimating the mean of every arm separately. Our lower bound is presented in Section 5. Lastly, in Section 6 we show simulations demonstrating that in practice, there is a large gap between the sample complexity of our algorithms and MEDIAN ELIMINATION. Additional literature review is in Appendix H. Running time and implementation issues are discussed in Appendix I.

## 2   SIMPLE APPROXIMATE BEST ARM ALGORITHM

In this section we present the Simple Approximate Best Arm (SABA) algorithm. SABA is a simplified version of the algorithm described in the next section. Its simplicity is achieved by making assumptions about the input to provably $(\epsilon, \delta)$-learn an a best arm. Namely, it assumes that $n \geq \max\{10^5, 1/\delta^4\}$ and that there is a unique $\epsilon$-best arm, i.e. all the arms in the input are $\epsilon$-far from $a^\star$. SABA is a concatenation of two procedures. The first is AGGRESSIVE ELIMINATION which is the main algorithmic idea behind this paper. The second is NAÏVE ELIMINATION which trivially samples all arms sufficiently many times and selecting the one with largest empirical mean.

## 2.1 Naïve Elimination

The following procedure is the naïve sampling approach to finding a best arm.

---
**Algorithm 1** NAÏVE ELIMINATION

---
**input** $\epsilon, \delta > 0$, arms $A$, noisy oracle for $\mu : A \to [0, 1]$
**output** arm in $A$ with largest empirical mean with $\frac{2}{\epsilon^2} \log \frac{|A|}{\delta}$ samples

---

The sample complexity of NAÏVE ELIMINATION is trivially $\frac{2|A|}{\epsilon^2} \log \frac{|A|}{\delta}$ and it returns an arm that is $\epsilon$-close to $a^\star$ with probability at least $1 - \delta$. We say that an arm $a \in A$ is $\eta$-**close** to $a' \in A$ if $\mu(a') - \mu(a) \leq \eta$ and $\eta$-**far** if $\mu(a') - \mu(a) > \eta$. One can obtain the approximation and confidence by bounding the likelihood of underestimating $a^\star$ and overestimating arms that are $\epsilon$-far from $a^\star$. For completeness we give full details in Appendix A. Throughout the paper we repeatedly use NAÏVE ELIMINATION with different values of $n$ and various approximation and confidence parameters. Also in Appendix A is the formal statement of the Hoffding Bound which is in much use in this work.

## 2.2 Agressive Elimination

The AGGRESSIVE ELIMINATION procedure that we introduce here iteratively discards arms with low empirical mean until reducing the total number of arms to $\frac{n^{3/4}}{2}$. To do so, in each round $i$ the procedure samples every arm $(i + 1)\frac{2}{\epsilon^2} \log \frac{1}{\delta}$ times and selects the $(\delta + \phi(n))$ fraction of arms whose sampled mean is highest into the next round. Intuitively, $\phi(n)$ is a small fraction s.t. the $(\delta + \phi(n))$ fraction of arms with largest sampled mean is likely to include $a^\star$. It is technically defined as:

$$\phi(n) = \sqrt{\frac{6 \log(n)}{n^{3/4}}}. \tag{1}$$

We will rely on this definition in Lemma 1 when analyzing the likelihood of $a^\star$ remaining in the final set of arms returned by the procedure. In particular, we bound the likelihood that $a^\star$ is underestimated and that other arms are overestimated. This definition of $\phi(n)$ is designed in such a way that we can later bound the likelihood that too many arms are overestimated, under certain assumptions.

The second term we define is $t(n)$ which is the number of iterations AGGRESSIVE ELIMINATION requires until reaching $\frac{n^{3/4}}{2}$ arms when we shrink the number of arms in each iteration by $\delta + \phi(n)$:

$$t(n) = \left\lceil \frac{\log n + 4 \log 2}{4 \log \left( \frac{1}{\delta + \phi(n)} \right)} \right\rceil. \tag{2}$$

Given these definitions we now formally describe and analyze AGGRESSIVE ELIMINATION below.

---
**Algorithm 2** AGGRESSIVE ELIMINATION

---
**input** $\epsilon, \delta > 0$, arms $A_0$, noisy oracle for $\mu : A_0 \to [0, 1]$
 1: **for** $i \in \{0, 1, 2, \ldots, t(n)\}$ **do**
 2:     apply $\ell_{i+1} = (i + 1) \left\lceil \frac{2}{\epsilon^2} \log \frac{1}{\delta} \right\rceil$ samples $\forall a \in A_i$
 3:     $A_{i+1} \leftarrow$ the $\lfloor |A_i| \times (\delta + \phi(n)) \rfloor$ best arms in $A_i$
 4: **end for**
**output** $A_{t(n)+1}$

---

**Sample complexity.** We will express the sample complexity of AGGRESSIVE ELIMINATION using $G(n, \delta)$ defined below. Importantly, $G(n, \delta)$ converges to 0 as $n$ grows and $\delta$ goes to 0:

$$G(n, \delta) = \sum_{i=1}^{t(n)} (\delta + \phi(n))^i (i + 1) \tag{3}$$

**Claim 1.** $\forall \epsilon, \delta \in [0,1]$, $n \geq 1$ *the sample complexity of* AGGRESSIVE ELIMINATION *is bounded by:*

$$\left(1 + G(n,\delta)\right) \times \left\lceil \frac{2n}{\epsilon^2} \log \frac{1}{\delta} \right\rceil.$$

*Proof.* Each iteration $i$ uses $\ell_{i+1} = (i+1)\left\lceil \frac{2}{\epsilon^2} \log \frac{1}{\delta} \right\rceil$ estimates on $|A_i| \leq n(\delta + \phi(n))^i$ arms. In total:

$$\sum_{i=0}^{t(n)} |A_i| \times \ell_{i+1} \leq \sum_{i=0}^{t(n)} n\left(\delta + \phi(n)\right)^i (i+1) \times \left\lceil \frac{2}{\epsilon^2} \log \frac{1}{\delta} \right\rceil = \left(1 + G(n,\delta)\right) \times \left\lceil \frac{2n}{\epsilon^2} \log \frac{1}{\delta} \right\rceil. \quad \square$$

Later in the paper we ignore the rounding of $\left\lceil \frac{2}{\epsilon^2} \log \frac{1}{\delta} \right\rceil$ and $\lfloor (\delta + \phi(n)) \rfloor$ when clear that the effect is negligible. The important takeaway is that the sample complexity of AGGRESSIVE ELIMINATION converges to $\frac{2n}{\epsilon^2} \log \frac{1}{\epsilon}$ as the number of arms grows and $\delta$ becomes small because $\lim_{n \to \infty, \delta \to 0} G(n,\delta) = 0$. Later in the paper we usually use non-asymptotic notion of $\delta$, and $G(n,\delta)$ is estimated more carefully.

**Likelihood of $a^\star$ surviving.** Next we analyze the likelihood of the best arm $a^\star$ to appear in the $\frac{n^{3/4}}{2}$ arms output of the AGGRESSIVE ELIMINATION procedure. We begin with a simple lemma that analyzes the likelihood of $|A_i| \cdot (\delta + \phi(n))$ arms – the number of arms with largest empirical mean we select in each iteration – to be $\frac{\epsilon}{2}$-overestimated. An arm $a \in A$ is $\eta$-**underestimated** if its empirical mean $\hat{\mu}(a)$ is evaluated to be less than $\eta$ of its true value, i.e. $\hat{\mu}(a) < \mu(a) - \eta$. An arm $a \in A$ is $\eta$-**overestimated** if $\hat{\mu}(a) > \mu(a) + \eta$. The proof is deferred to Appendix A.

**Lemma 1.** *For every iteration $i \in \{0, 1, \ldots, t(n)\}$ of* AGGRESSIVE ELIMINATION *the probability that more than $|A_i| \cdot (\delta + \phi(n))$ arms are $\frac{\epsilon}{2}$-overestimated at iteration $i$ is smaller than $\frac{1}{n^6}$.*

The main idea that we now show is that with sufficient probability in every round, $a^\star$ is not $\frac{\epsilon}{2}$-underestimated and sufficiently few $\epsilon$-far arms are $\frac{\epsilon}{2}$-overestimated. Showing this implies that in every round $a^\star$ is one of the arms with highest empirical mean and selected to the next round.

**Claim 2.** *Suppose the $\epsilon$-best arm $a^\star$ is unique, i.e. all arms are $\epsilon$-far from $a^\star$. Then, the likelihood that* AGGRESSIVE ELIMINATION *returns a set of arms $A_{t(n)+1}$ that does not contain $a^\star$ is at most:*

$$\delta\left(\frac{1}{1-\delta}\right) + \left(n^5 \log\left(\frac{1}{\delta + \phi(n)}\right)\right)^{-1}.$$

*Proof.* We will analyze the likelihood that $a^\star$ is not selected into $A_{i+1}$, given that it is in $A_i$, for every $i \in \{0, 1, \ldots, t(n)\}$. In every iteration $i$ we can bound the likelihood of $a^\star$ being $\frac{\epsilon}{2}$-underestimated:

$$\Pr\left[\hat{\mu}(a^\star) < \mu(a^\star) - \frac{\epsilon}{2}\right] \leq e^{\frac{-\epsilon^2 \ell_{i+1}}{2}} = e^{-(i+1)\log \frac{1}{\delta}} = \delta^{i+1}$$

By definition of AGGRESSIVE ELIMINATION $a^\star$ is not in $A_{i+1}$ only if there are at least $|A_i|(\delta + \phi(n))$ arms in $A_i$ whose empirical mean is higher than that of $a^\star$. By the assumption of the claim, we know that all other arms are $\epsilon$-far from $a^\star$. If $a^\star$ does not survive to the next round it is because it was $\frac{\epsilon}{2}$-underestimated or at least $|A_i|(\delta + \phi(n))$ arms were $\frac{\epsilon}{2}$-overestimated. By Lemma 1 we know that the likelihood of more than $|A_i|(\delta + \phi(n))$ arms to be $\frac{\epsilon}{2}$-overestimated is $n^{-6}$. Thus, by a union bound, in every iteration $i \in \{0, 1, \ldots, t(n)\}$ the likelihood of discarding $a^\star$ is at most $\delta^{i+1} + n^{-6}$. The likelihood that $a^\star$ does not survive the last elimination is at most:

$$\sum_{i=0}^{t(n)} \left(\delta^{i+1} + \frac{1}{n^6}\right) = \left(\sum_{i=0}^{t(n)} \delta^{i+1}\right) + \frac{t(n)}{n^6} < \delta\left(\frac{1}{1-\delta}\right) + \frac{1}{n^5\left(\log \frac{1}{\delta+\phi(n)}\right)}. \quad \square$$

The main takeaway is that when $n$ is sufficiently large as a function of $\delta$, there is a high probability that $a^\star$ is in the set of arms returned by the procedure when the rest of arms are $\epsilon$-far from $a^\star$.

## 2.3 A Simple Algorithm under Favorable Conditions

At this point learning a best arm under favorable conditions seems rather straightforward: we implement AGGRESSIVE ELIMINATION and then run NAÏVE ELIMINATION on the remaining set of $\frac{n^{3/4}}{2}$ arms. We present the algorithm formally below and give details of the analysis in Appendix A.

---

**Algorithm 3** SIMPLE APPROXIMATE BEST ARM

---

**input** arms $A$, $\epsilon, \delta > 0$, noisy oracle for $\mu : A \rightarrow [0, 1]$
  1: $A_T \leftarrow$ AGGRESSIVE ELIMINATION$(A, \epsilon, \frac{\delta}{2})$
**output** NAÏVE ELIMINATION$(A_T, \epsilon, \frac{\delta}{e})$

---

**Claim 3.** *Assume that there is a unique $\epsilon$-best arm in $A$. Then $\forall \delta \leq 0.05$ and $n \geq \max\{1/\delta^4, 10^5\}$, SABA $(\epsilon, \delta)$-learns a best arm with sample complexity $\frac{4n}{\epsilon^2} \log \frac{1}{\delta}$.*

# 3 APPROXIMATE BEST ARM ALGORITHM

In this section we present the Approximate Best Arm (ABA) algorithm which is a modification of SABA. We first discuss how to remove the assumptions SABA makes and then describe the algorithm.

**Removing $n \geq \max\{1/\delta^4, 10^5\}$ assumption.** When we seek a bound that holds for any $n$ (i.e. not an asymptotic bound for $n \rightarrow \infty$) we avoid this assumption by simply running NAÏVE ELIMINATION when the parameters do not respect these conditions. It is easy to verify that when $n < 1/\delta^4$ or $n < 10^5$ and $\delta < 0.05$ we can $(\epsilon, \delta)$-learn a best arm by running NAÏVE ELIMINATION$(A, \epsilon, \delta)$ and the sample complexity is then $\frac{10n}{\epsilon^2} \log \frac{1}{\delta}$. When we analyze the asymptotic result in Section 3.1, we'll show a different modification of the algorithm that doesn't require running NAÏVE ELIMINATION.

**Removing the unique $\epsilon$-best arm assumption.** To avoid this assumption we will slightly decrease $\epsilon$ and apply AGGRESSIVE ELIMINATION with $\epsilon_0 = \alpha \cdot \epsilon$ using $\alpha \in [0, 1]$ that we later define. In addition, we will select a random set of size $\frac{n^{7/8}}{2}$. Together, this guarantees that we are likely to have an arm that is $\epsilon_0$-close to $a^\star$, either in the random set or the output of AGGRESSIVE ELIMINATION:

- We prove a claim similar to Claim 2 but under weaker conditions. Specifically we show that as long as there are fewer than $\frac{n^{3/8}}{4}$ arms that are $\epsilon_0$-close to $a^\star$, then with sufficient confidence $a^\star$ will be one of the arms returned in $A_T$;

- Otherwise, there are more than $\frac{n^{3/8}}{4}$ arms that are $\epsilon_0$-close to $a^\star$ and one will surface with overwhelming probability (as a function of $n$) in a random set $R$ of size $\frac{n^{7/8}}{2}$.

Consequently, it is very likely that there is an $\epsilon_0$-close arm either in $A_T$ or in the random set $R$ (or both) and running NAÏVE ELIMINATION with appropriate parameters on $A_T \cup R$ will return an $\epsilon$-best arm with probability at least $1 - \delta$.

**The algorithm.** The Approximate Best Arm (ABA) algorithm described below is a modification of SABA that incorporates the modifications discussed above.

---

**Algorithm 4** APPROXIMATE BEST ARM

---

**input** arms $A$, $\alpha, \epsilon, \delta > 0$, noisy oracle for $\mu : A \rightarrow [0, 1]$
  1: **initialize** $R \leftarrow \frac{n^{7/8}}{2}$ arms selected u.a.r.
  2: **if** $n < \max\{10^5, \delta^{-4}\}$ **output** NAÏVE ELIMINATION$(A, \epsilon, \delta)$
  3: $A_T \leftarrow$ AGRESSIVE ELIMINATION$(A, \alpha \cdot \epsilon, \frac{\delta}{2})$
**output** NAÏVE ELIMINATION$(A_T \cup R, (1 - \alpha)\epsilon, \frac{\delta}{e})$

---

We first generalize Claim 2 for the case in which there isn't necessarily a unique $\epsilon$-best arm $a^\star$ but rather at most $\frac{n^{3/8}}{4}$ arms that are $\epsilon$-close to $a^\star$. The proof is similar and deferred to Appendix B.

**Claim 4.** *Suppose that there are at most $\frac{n^{3/8}}{4}$ arms that are $\epsilon$-close to $a^\star$ in $A$ and the rest are $\epsilon$-far. Then, the likelihood that* AGGRESSIVE ELIMINATION$(A, \epsilon, \delta)$ *returns a set of arms $A_{t(n)+1}$ that does not contain $a^\star \notin A_{t(n)+1}$ is at most:*

$$\delta \left( \frac{1}{1 - \delta} \right) + \left( n \log \left( \frac{1}{\delta + \phi(n)} \right) \right)^{-1}.$$

We now state the approximation and confidence of ABA. We provide proof sketches that are devoid of some of the calculations, and give full proofs in Appendix B.

**Lemma 2.** *For any $\delta \leq 0.05$* ABA *initialized with $\alpha = 1 - 1/e$ returns an $\epsilon$-best arm w.p. $\geq 1 - \delta$.*

*Sketch.* If $n < \max\{1/\delta^4, 10^5\}$ we invoke NAÏVE ELIMINATION which is guaranteed to return an $\epsilon$-best arm with confidence $1 - \delta$. Otherwise, we assume that $n \geq \max\{1/\delta^4, 10^5\}$ and we can analyze the performance of AGGRESSIVE ELIMINATION invoked with $\alpha\epsilon$ and $\delta' = \delta/2$.

In the case that there are at most $\frac{n^{3/8}}{4}$ arms that are $\alpha\epsilon$-close to $a^\star$ then according to Claim 4 AGGRESSIVE ELIMINATION invoked with $\alpha\epsilon$ and $\delta' = \delta/2$ will not include $a^\star$ in $A_T$ w.p. at most:

$$\delta' \left( \frac{1}{1 - \delta'} \right) + \left( n \log \left( \frac{1}{\delta' + \phi(n)} \right) \right)^{-1} < (1 - 1/e)\delta$$

Conditioned on $a^\star \in A_T$ the likelihood that NAÏVE ELIMINATION on $A_T \cup R$ with approximation $(1 - \alpha)\epsilon < \epsilon$ does not return an $\epsilon$-best arm is at most $\delta/e$. Thus, if there are at most $\frac{n^{3/8}}{4}$ arms that are $\alpha\epsilon$-close to $a^\star$ the algorithm terminates with an $\epsilon$-best arm with probability at least $1 - \delta$.

Otherwise, there are at least $\frac{n^{3/8}}{4}$ arms that are $\alpha\epsilon$-close to $a^\star$. Since we select arms to $R$ u.a.r. and $|R| = \frac{n^{7/8}}{2}$ the likelihood of not having any arms that are $\alpha\epsilon$-close in $R$ is smaller than $(1 - 1/e)\delta$. Let $\tilde{a}$ be an arm that is $\alpha\epsilon$-close to $a^\star$ in $R$. When we run NAÏVE ELIMINATION with approximation $(1 - \alpha)\epsilon$ and $\delta/e$, we are guaranteed that with probability at least $1 - \delta/e$ no arm that is $\epsilon$-far from $a^\star$ will have empirical mean higher than that of $\tilde{a}$. Since $\tilde{a}$ is $\alpha\epsilon$-close to $a^\star$ and $\alpha < 1$ this implies that the algorithm returns an arm that is at least $\epsilon$-close to $a^\star$ w.p. at least $1 - \delta$ in this case as well. $\square$

**Theorem 1.** *For any $\delta \leq 0.05$* ABA *initialized with $\alpha = 1 - 1/e$ returns an $\epsilon$-best arm w.p. at least $1 - \delta$ using total number of samples of at most: $18 \times \frac{n}{\epsilon^2} \log \frac{1}{\delta}$.*

*Sketch.* If $n < 1/\delta^4$ or $n < 10^5$ we invoke NAÏVE ELIMINATION and its sample complexity is $\frac{10n}{\epsilon^2} \log \frac{1}{\delta}$. According to Claim 1 the sample complexity of AGGRESSIVE ELIMINATION with approximation $\alpha\epsilon$ and confidence $\delta^{1+c}$ the sample complexity is:

$$\frac{1}{\alpha^2} \left( \frac{2n}{\epsilon^2} \log \frac{1}{\delta} \left( (1 + c) \left( 1 + G(n, \delta^{1+c}) \right) \right) \right) \tag{4}$$

For any $\delta < 0.05$ we have that $\delta^{1+c} < \delta/2$ for $c = 1/4$. Thus, since we ran AGGRESSIVE ELIMINATION with confidence $\delta/2$ and $\alpha = 1 - 1/e$ the sample complexity is at most:

$$\frac{1}{\alpha^2} \left( \frac{2n}{\epsilon^2} \log \frac{1}{\delta} \left( (1 + c) \left( 1 + G(n, \delta^{1+c}) \right) \right) \right) < 8 \left( \frac{n}{\epsilon^2} \log \frac{1}{\delta} \right)$$

For the sample complexity of the NAÏVE ELIMINATION notice that it is applied on $B = A_T \cup R$. Since $\alpha = 1 - 1/e$ and $|B| = \frac{n^{3/4}}{2} + \frac{n^{7/8}}{2}$, the sample complexity of NAÏVE ELIMINATION is:

$$\frac{1}{(1 - \alpha)^2} \left( \frac{2|B|}{\epsilon^2} \log \left( \frac{|B|}{\delta} \right) \right) < 10 \left( \frac{n}{\epsilon^2} \log \frac{1}{\delta} \right) \tag{5}$$

Therefore, the sample complexity of AGGRESSIVE ELIMINATION and NAÏVE ELIMINATION is $18 \times \frac{n}{\epsilon^2} \log \frac{1}{\delta}$ and the total sample complexity is bounded by $\frac{18n}{\epsilon^2} \log \frac{1}{\delta}$. $\square$

## 3.1 Asymptotic Sample Complexity

In our exposition of ABA above, we fixed some parameters to show that it achieves low sample complexity for any value of $n$. This sample complexity is due (1) NAÏVE ELIMINATION to ensure that $n > \max\{10^5, 1/\delta^4\}$ and (2) a convex combination of AGGRESSIVE ELIMINATION and NAÏVE ELIMINATION applied on a sublinear number of arms $A_T \cup R$. Intuitively, to remove (1), if we allow $n$ grow, we can remove the NAÏVE ELIMINATION procedure. For (2) Recall from Claim 1 that the sample complexity of AGGRESSIVE ELIMINATION is:

$$\left(1 + G(n, \delta)\right) \times \frac{2n}{\epsilon^2} \log \frac{1}{\delta}.$$

Since $\lim_{n \to \infty, \delta \to 0} G(n, \delta) = 0$, this converges to sample complexity of $\frac{2n}{\epsilon^2} \log \frac{1}{\delta}$. What remains is the NAÏVE ELIMINATION applied on a sublinear number of arms $A_T \cup R$. Intuitively, since the number of arms is sublinear in $n$, as $n$ grows the sample complexity converges to 0. We elaborate on the asymptotic results in Appendix B.1 and prove the following theorem.

**Theorem 2.** *For any $\lambda > 0$ there exist $\delta_0$ and $n_0$ s.t. for any $\delta < \delta_0$ and $n \geq n_0$, ABA $(\epsilon, \delta)$-learns a best arm with sample complexity at most:* $\left(2 + \lambda\right) \frac{n}{\epsilon^2} \log \frac{1}{\delta}$.

# 4 APPROXIMATE BEST ARM BY HOEFFDING

We now describe the Approximate Best Arm Likelihood Estimation (ABALEH) algorithm. This algorithm is a variant of ABA which achieves a sample complexity that is arbitrarily close to that of $(\epsilon, \delta)$-learning the mean of every arm. Unlike ABA here we must assume that $n \geq 1/\delta$.

In this algorithm, we want to circumvent the barrier of $2 \times \frac{n}{\epsilon^2} \log \frac{1}{\delta}$ of ABA and get to the complexity of $(1 + \lambda) \times \frac{n}{2\epsilon^2} \log \frac{1}{\delta}$ for arbitrarily small $\lambda > 0$. The main idea is that to determine that one arm is better than the other (assuming they are $\epsilon$-far) it is also possible to estimate one of them to accuracy $(1 - \zeta)\epsilon$ and the other to accuracy $\zeta\epsilon$ for $\zeta > 0$ that we choose later. We sample each arm $(1 + \frac{\lambda}{2}) \frac{1}{2\epsilon^2} \log \frac{1}{\delta}$ times, but in the analysis we apply a different Hoeffding bound per arm:

1. For the best arm, in the analysis we apply a Hoeffding bound with accuracy $(1 - \zeta)\epsilon$ and failure probability $\ll \delta$. This ensures the best arm is approximated up to almost $\epsilon$;

2. For any other arm we apply Hoeffding with accuracy $\zeta\epsilon$, and failure probability $\gg \delta$. The number of samples on each arm is still bounded by $(1 + \frac{\lambda}{2}) \frac{1}{2\epsilon^2} \log \frac{1}{\delta}$, as we pay for the additional accuracy with higher failure probability. This is where we need $\delta$ to be small.

Note that we do not assume the algorithm knows which is the best arm, but the analysis can apply different theorems to different arms. Since there are $n - 1$ arms which are not the best arm, and $n$ is large, we can know how many of them failed the Hoeffding bound. As long as this number is not too large (say $0.001n$) we can be sure that the best arm moves the next stage with high probability. To choose $\zeta$, notice that if there were only two arms, it would be wise to choose $\zeta = 1/2$, but for an arbitrary number of arms we use a smaller $\zeta$ and take $\zeta = 1 - (1 - \frac{\lambda}{16})\sqrt{1 - \frac{\lambda}{8}}$ where $\lambda$ is a parameter of the algorithm. We defer the proofs to Appendix C.

---

**Algorithm 5** APPROXIMATE BEST ARM LIKELIHOOD ESTIMATION BY HOEFFDING

---

**input** $\epsilon, \delta, \lambda \in (0, 1)$, arms $A$, noisy oracle for $\mu : A \to [0, 1]$

1: $\alpha \leftarrow \sqrt{1 - \frac{\lambda}{8}}$

2: $R \leftarrow$ a random set of $n^{3/4}$ arms

3: apply $(1 + \frac{\lambda}{2})(\frac{1}{2\epsilon^2} \log \frac{1}{\delta})$ samples $\forall a \in A$

4: $A_0 \leftarrow$ the $\frac{\lambda n}{50}$ highest estimated arms in $A$

5: $A_T \leftarrow$ AGRESSIVE ELIMINATION$(A_0, \epsilon\alpha, \frac{\delta}{4})$

**output** NAÏVE ELIMINATION$(A_T \cup R, (1-\alpha)\epsilon, \frac{\delta}{4})$

---

**Lemma 3.** *Suppose $\lambda < 1$, $\delta \le \delta_0$ where $\delta_0$ is the solution to $\frac{\lambda}{100} = \delta_0^{\lambda^2/256}$, and $n > 1/\delta$. If there are at most $n^{2/3}$ arms which are $\alpha\epsilon$-close to $a^\star$ then w.p. at least $1 - \frac{\delta}{2}$ we have that $a^\star$ is one of the $\frac{\lambda n}{50}$ highest estimated arms in A.*

Given Lemma 3, the proof now follows in a similar manner to previous proofs by bounding the sample complexity and approximation and confidence of all sub procedures.

**Theorem 3.** *For any given $\lambda < 1$ there is a $\delta_0$ s.t. for any $\delta < \delta_0$ and $n > 1/\delta$ ABALEH $(\epsilon, \delta)$-learns a best arm with sample complexity at most:*

$$\left(1 + \lambda\right) \frac{n}{2\epsilon^2} \log \frac{1}{\delta}.$$

## 5   LOWER BOUND

We now consider the family of *elimination algorithms* denoted $\mathcal{F}$ and defined as follows. An algorithm is in $\mathcal{F}$ if it begins when $S = A$ is the set of all possible arms and then: (i) pulls each arm in $S$ once (ii) eliminates some of the arms in $S$, and (iii) if $|S| = 1$ terminate, else, go back to (i).

Since best arm algorithms have very little degrees of freedom many of them are elimination algorithms. Essentially, the only limitation here is that the algorithm's decisions are irrevocable: if the algorithm considers an arm to be suboptimal and discards it from consideration, it cannot revoke and decision and consider the arm again. We defer the full proof of the lower bound to Appendix D.

**Theorem 4.** *For every $\beta > 0$ there exist $\epsilon_0, \delta_0$ such that for any algorithm in $\mathcal{F}$ which finds an $\epsilon$ best arm with success probability $1 - \delta$ where $\epsilon < \epsilon_0$, $\delta < \delta_0$, there exist $n_0$ such that if $n > n_0$, the algorithm requires at least $\left(\frac{1}{2} - \beta\right) \frac{n}{\epsilon^2} \log \frac{1}{\delta}$ queries.*

## 6   Experiments

To illustrate the efficiency of the algorithms we conducted a simple numerical experiment. A reasonable concern may be that while our results suggest a dramatic improvement over the sample complexity of MEDIAN ELIMINATION this improvement may only be due to tighter analysis. In this section we rule out this possibility by experimentally comparing the actual sample complexity (not analysis) of our algorithms (SABA, ABA and ABALEH) with MEDIAN ELIMINATION and NAÏVE ELIMINATION. Note that all algorithms are guaranteed to $(\epsilon, \delta)$-learn the best arm, and thus our interest is in their sample complexity. Since our algorithms relative sample complexity improves as $n$ grows we were interested in observing this improvement emprically. Due to lack of space, we differ the full results to Appendix F. In short, we show that for reasonable input sizes, ABALEH have a sample complexity which is 100 times more efficient than MEDIAN ELIMINATION.

#### Acknowledgements
Avinatan Hassidim - This work is supported by ISF and by BSF grant number 2016015.
Ron Kupfer - This project has received funding from the European Research Council (ERC) under the European Union's Horizon 2020 research and innovation program (grant agreement No. 740282).
Yaron Singer - This research was supported by BSF grant 2014389, NSF grant CAREER CCF-1452961, NSF USICCS proposal 1540428, Google research award, and a Facebook research award.

## 7   Broader Impact

Learning a best arm is a fundamental, well-studied, problem largely because it captures the most basic experimental question: given $n$ treatments, each with a stochastic outcome, which one is best? Cancer treatment, drug discovery, gene detection, manufacturing quality assurance, financial fraud detection, spam detection, software testing, are all examples of direct applications of learning a best arm. Providing dramatically faster algorithms for these applications without compromising on guarantees will impact areas well outside machine learning. Specifically, this work provides an algorithm that is 6000 times faster than the state-of-the-art. In addition to asymptotic bounds that converge as the number of arms grows to what we conjecture is the optimal sample complexity, we provide dramatic speedups for any number of arms. The result is an extremely efficient simple algorithm for learning

a best arm with strong theoretical guarantees that can be used across all applications of learning a best arm. The simplicity and speed of the algorithms presented here are such that any practitioner can implement them and accelerate their experimental setup immediately. We trust that we will see immediate action across a broad set of application domains.

## Footnotes

[1]All the results in this paper can be generalized for any sub-Gaussian distribution as discuss in Appendix E.

[2] In particular, our algorithms use the naive elimination strategy when $n < 10^5$. For MEDIAN ELIMINATION the naive strategy has better sample complexity for any $n < 2^{1500}$.

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
