[Supplementary Material · best_arm_nips20.pdf]

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

[3]We verified that changing $\epsilon$ has no effect on the ratio of the number of samples required by the algorithms.

[4]i.e. the trivial elimination procedure which given $k$ arms, samples each arm $\frac{2}{\epsilon^2} \log \frac{k}{\delta}$ times and selects the arm with highest empirical mean. It is easy to see that this is an $(\epsilon, \delta)$-learning strategy.

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

## A    Simple Approximate Best Arm Algorithm

**Theorem 5** (Hoeffding bound). *Let $X_1, ..., X_m$ be independent random variables bounded by the interval $[0,1]$, $\bar{X} = \frac{1}{m}\sum_{i=1}^{m} X_m$ and $m = \frac{1}{2\epsilon^2}\log\frac{1}{\delta}$, then $\Pr(\bar{X} - \mathbb{E}[\bar{X}] \geq \epsilon) \leq \delta$.*

**Claim 5.** *The sample complexity of* NAÏVE ELIMINATION *is $\frac{2|A|}{\epsilon^2}\log\frac{|A|}{\delta}$ and it returns an arm that is $\epsilon$-close to $a^\star$ with probability at least $1 - \delta$.*

*Proof.* To see this, suppose that $a^\star$ is not returned by NAÏVE ELIMINATION. If another arm is returned that is $\epsilon$-close to $a^\star$ then we are done. Otherwise, assume that NAÏVE ELIMINATION returns an arm $a$ that is $\epsilon$-far. Since any arm is sampled $\frac{1}{2(\epsilon/2)^2}\log\frac{1}{\delta}$ times, by the Hoeffding bound we know that the likelihood of either $\frac{\epsilon}{2}$-underestimating $a^\star$ or $\frac{\epsilon}{2}$-overestimating an $\epsilon$-far arm is $\frac{\delta}{|A|}$. There are at most $|A| - 1$ arms that are $\epsilon$-far from $a^\star$. By a union bound, $a^\star$ is not $\frac{\epsilon}{2}$ underestimated and none of the $\epsilon$-far arms are $\frac{\epsilon}{2}$-overestimated w.p. at least $1 - \delta$. Thus $a^\star$ has larger empirical mean than any of the $\epsilon$-far arms, implying that the procedure returns an $\epsilon$-best arm w.p. at least $1 - \delta$.  □

**Lemma 1.** *For every iteration $i \in \{0, 1, \ldots, t(n)\}$ of* AGGRESSIVE ELIMINATION *the probability that more than $|A_i| \cdot (\delta + \phi(n))$ arms are $\frac{\epsilon}{2}$-overestimated at iteration $i$ is smaller than $\frac{1}{n^6}$.*

*Proof.* In every iteration $i \in \{0, 1, \ldots, t(n)\}$ the likelihood of arm $a \in A$ being $\frac{\epsilon}{2}$-overestimated is:

$$\Pr\left[\hat{\mu}(a) > \mu(a) + \frac{\epsilon}{2}\right] \leq e^{\frac{-\epsilon^2 \ell_{i+1}}{2}} = e^{-(i+1)\log\frac{1}{\delta}} = \delta^{i+1}$$

Therefore, in expectation, there are $|A_i| \cdot \delta^{i+1}$ arms that are $\frac{\epsilon}{2}$-overestimated. Let $X_a$ denote the random variable that indicates whether arm $a$ is $\frac{\epsilon}{2}$ overestimated, $X = \sum_{a \in A_i} X_a$ and $\hat{X}$ be the number of arms that are $\frac{\epsilon}{2}$-overestimated at iteration $i$. Again, by Hoeffding, the likelihood of more than $|A_i|(\delta + \phi(n))$ being $\frac{\epsilon}{2}$-overestimated:

$$\Pr\left[|A_i| \cdot (\delta + \phi(n)) \text{ arms } \frac{\epsilon}{2}\text{-overestimated}\right] = \Pr\left[\hat{X} - \mathbb{E}[X] \geq (\delta + \phi(n))|A_i| - \mathbb{E}[X]\right] \quad (6)$$

$$= \Pr\left[\hat{X} - \mathbb{E}[X] \geq (\delta - \delta^{i+1} + \phi(n))|A_i|\right] \quad (7)$$

$$\leq \Pr\left[\hat{X} - \mathbb{E}[X] \geq \phi(n)|A_i|\right] \quad (8)$$

$$\leq \exp\left(-2\phi(n)^2|A_i|\right) \quad (9)$$

$$\leq \exp\left(-\phi(n)^2 n^{3/4}\right) \quad (10)$$

$$= \frac{1}{n^6} \quad (11)$$

In (7) we use the fact that $\mathbb{E}[X] = |A_i| \cdot \delta^{i+1}$, in (8) we used the fact that $\delta \leq 1$, in (10) we used the fact that there are at least $\frac{n^{3/4}}{4}$ arms in $A_i$, and in (30) we used the definition of $\phi(n)$ in (1).  □

**Claim 3.** *Assume that there is a unique $\epsilon$-best arm in $A$. Then $\forall \delta \leq 0.05$ and $n \geq \max\{1/\delta^4, 10^5\}$,* SABA $(\epsilon, \delta)$*-learns a best arm with sample complexity $\frac{4n}{\epsilon^2}\log\frac{1}{\delta}$.*

*Proof.* The proof follows from the sample complexity and approximation and confidence of AGGRESSIVE ELIMINATION and NAÏVE ELIMINATION. The sample complexity is the total number of samples required to implement AGGRESSIVE ELIMINATION with $\delta/2$ and NAÏVE ELIMINATION on the remaining arms with $\delta/e$. A convenient way to express the sample complexity of AGGRESSIVE ELIMINATION is to use a constant $c$ for which $\delta^{1+c} = \delta/2$. The sample complexity of AGGRESSIVE ELIMINATION with $\delta^{1+c}$ is:

$$(1 + c)\left(1 + G(n, \delta^{1+c})\right) \times \frac{2n}{\epsilon^2}\log\frac{1}{\delta} \quad (12)$$

In our case, if we assume that $\delta < 0.05$ then for $c = 1/4$ we get that $\delta^{1+c} < \delta/2$.

For NAÏVE ELIMINATION executed on $\frac{n^{3/4}}{2}$ arms with $\delta/e$ the sample complexity is:

$$\left( \frac{1}{2n^{\frac{1}{4}}} \left( 1 + \frac{3 \log n + 4}{4 \log \frac{1}{\delta}} \right) \right) \times \frac{2n}{\epsilon^2} \log \frac{1}{\delta} \tag{13}$$

For $n \geq 10^5$ and $\delta < 0.05$ the sample complexity of SABA is $(12) + (13) < \frac{4n}{\epsilon^2} \log \frac{1}{\delta}$.

In terms of approximation and confidence, for $n \geq 10^5$ then $\phi(n) < 0.12$ and for $\delta < 0.05$ we get $\log(\frac{1}{\delta' + \phi(n)}) > 1$. Applying AGGRESSIVE ELIMINATION on $\delta' = \delta/2$ when $\delta \leq 0.05$ implies that $a^\star$ is not in $A_T$ w.p. at most:

$$\delta' \left( \frac{1}{1 - \delta'} \right) + \frac{1}{n^5 \log(\frac{1}{\delta' + \phi(n)})} < \frac{20\delta}{39} + \frac{1}{n^5} < \left( 1 - \frac{1}{e} \right) \delta$$

Finally, assuming that $a^\star$ is in $A_T$ then the probability it is not returned by NAÏVE ELIMINATION is at most $\delta/e$. By union bound, the likelihood that $a^\star$ is either not in $A_{t+1}$ or not selected by NAÏVE ELIMINATION is at most $\delta$. $\qquad\square$

## B Approximate Best Arm Algorithm

**Claim 4.** *Suppose that there are at most $\frac{n^{3/8}}{4}$ arms that are $\epsilon$-close to $a^\star$ in $A$ and the rest are $\epsilon$-far. Then, the likelihood that AGGRESSIVE ELIMINATION$(A, \epsilon, \delta)$ returns a set of arms $A_{t(n)+1}$ that does not contain $a^\star \notin A_{t(n)+1}$ is at most:*

$$\delta \left( \frac{1}{1 - \delta} \right) + \left( n \log \left( \frac{1}{\delta + \phi(n)} \right) \right)^{-1}.$$

*Proof.* Since there are at most $\frac{n^{3/8}}{4}$ arms that are $\epsilon$-close, we know that in every iteration $i \in \{0, 1, \ldots, t(n)\}$ there are at least $|A_i| - \frac{n^{3/8}}{4}$ that are $\epsilon$-far from $a^\star$. In the worst case, in every iteration every one of the $\epsilon$-close arms is overestimated in such a way that its empirical mean is larger than that of $a^\star$. In this case, the only way that $a^\star$ is not included in round $A_{i+1}$ is if $a^\star$ is $\frac{\epsilon}{2}$-underestimated and there are more than $|A_i|(\delta + \phi(n)) - \frac{n^{3/8}}{4}$ arms that are $\epsilon$-far that are $\frac{\epsilon}{2}$ overestimated.

As in the proof of Claim 2 using the Hoeffding bound we know that the likelihood of $a^\star$ being $\frac{\epsilon}{2}$-underestimated is at most $\delta^{i+1}$. The likelihood of an arm being $\frac{\epsilon}{2}$-overestimated is at most $\delta^{i+1}$ and in expectation there are $|A_i|\delta^{i+1}$ arms that are $\frac{\epsilon}{2}$-overestimated in every iteration $i$. Let $X_a$ denote the random variable that indicates whether arm $a$ is $\frac{\epsilon}{2}$-overestimated, $X = \sum_{a \in A_i} X_a$ and $\hat{X}$ be the number of arms that are $\frac{\epsilon}{2}$-overestimated at iteration $i$. Again, by Hoeffding, the likelihood of more than $|A_i|(\delta + \phi(n)) - \frac{n^{3/8}}{4}$ being $\frac{\epsilon}{2}$-overestimated:

$$\Pr\left[\hat{X} \geq (\delta + \phi(n))|A_i| - \frac{n^{3/8}}{4}\right] = \Pr\left[\hat{X} - \mathbb{E}[X] \geq (\delta + \phi(n))|A_i| - \mathbb{E}[X] - \frac{n^{3/8}}{4}\right] \quad (14)$$

$$= \Pr\left[\hat{X} - \mathbb{E}[X] \geq (\delta - \delta^{i+1} + \phi(n))|A_i| - \frac{n^{3/8}}{4}\right] \quad (15)$$

$$\leq \Pr\left[\hat{X} - \mathbb{E}[X] \geq \phi(n)\left(|A_i| - \frac{n^{3/8}}{4\phi(n)}\right)\right] \quad (16)$$

$$\leq \Pr\left[\hat{X} - \mathbb{E}[X] \geq \phi(n)\left(|A_i| - \frac{n^{3/4}}{12}\right)\right] \quad (17)$$

$$\leq \Pr\left[\hat{X} - \mathbb{E}[X] \geq \phi(n)\left(\frac{2}{3}|A_i|\right)\right] \quad (18)$$

$$\leq \exp\left(-\frac{2 \cdot 4\phi(n)^2|A_i|}{9}\right) \quad (19)$$

$$\leq \exp\left(-\frac{2\phi(n)^2 n^{3/4}}{9}\right) \quad (20)$$

$$= \frac{1}{n^{4/3}} \quad (21)$$

In (15) we use the fact that $\mathbb{E}[X] = |A_i| \cdot \delta^{i+1}$, in (16) we used the fact that $\delta \leq 1$, in (17) we used the fact that $\phi(n) > 3n^{-3/8}$ for $n \geq 5$, in eq:delta4 and (20) we used the fact that there are at least $\frac{n^{3/4}}{4}$ arms in $A_i$, and in (21) we used the definition of $\phi(n)$ in (1).

Having calculated the likelihood that $a^\star$ is $\frac{\epsilon}{2}$-underestimated to be $\delta^{i+1}$ and the likelihood that there are at least $|A_i|(\delta + \phi(n)) - \frac{n^{3/8}}{4}$ arms that are $\frac{\epsilon}{2}$-overestimated, by a union bound, in every iteration $i \in \{0, 1, \ldots, t(n)\}$ the likelihood of discarding $a^\star$ is at most:

$$\delta^{i+1} + \frac{1}{n^{4/3}}$$

Taking a union bound over the likelihood that $a^\star$ is discarded in every iteration $i \in \{0, 1, \ldots, t(n)\}$ we get that the likelihood that $a^\star$ does not survive the last elimination step is at most:

$$\sum_{i=0}^{t(n)} \left(\delta^{i+1} + \frac{1}{n^{4/3}}\right) = \left(\sum_{i=0}^{t(n)} \delta^{i+1}\right) + \frac{t(n)}{n^{4/3}} < \delta\left(\frac{1}{1-\delta}\right) + \frac{1}{n\left(\log\frac{1}{\delta+\phi(n)}\right)}.$$

$\square$

**Lemma 2.** *For any $\delta \leq 0.05$ ABA initialized with $\alpha = 1 - 1/e$ returns an $\epsilon$-best arm w.p. $\geq 1 - \delta$.*

*Proof.* If $n < \max\{1/\delta^4, 10^5\}$ we invoke NAÏVE ELIMINATION which is guaranteed to return an $\epsilon$-best with confidence $1 - \delta$. Otherwise, we assume that $n \geq \max\{1/\delta^4, 10^5\}$ and we can analyze the performance of AGGRESSIVE ELIMINATION invoked with $\alpha\epsilon$ and $\delta' = \delta/2$.

In the case that there are at most $\frac{n^{3/8}}{4}$ arms that are $\alpha\epsilon$-close to $a^\star$ then according to Claim 4 AGGRESSIVE ELIMINATION invoked with $\alpha\epsilon$ and $\delta' = \delta/2$ will include $a^\star$ in $A_T$ w.p. at least :

$$\delta'\left(\frac{1}{1-\delta'}\right) + \left(n\log\left(\frac{1}{\delta'+\phi(n)}\right)\right)^{-1}$$

When $\delta' = \frac{\delta}{2}$ and $\delta < 0.05$ we have that $\delta'\left(\frac{1}{1-\delta'}\right) < \frac{20}{39}\delta$. When $\delta < 0.05$ then $\log\left(\frac{1}{\delta'+\phi(n)}\right) > 1$ and since $n \geq 1/\delta^4$ we have that

$$\left(n\log\left(\frac{1}{\delta'+\phi(n)}\right)\right)^{-1} < \delta^4.$$

Together we have that the likelihood that $a^\star$ is not in $A_T$ returned by AGGRESSIVE ELIMINATION is:

$$\delta'\left(\frac{1}{1-\delta'}\right) + \left(n\log\left(\frac{1}{\delta'+\phi(n)}\right)\right)^{-1} < (1-1/e)\delta$$

Conditioned on $a^\star \in A_T$ the likelihood that NAÏVE ELIMINATION on $A_T \cup R$ with approximation $(1-\alpha)\epsilon < \epsilon$ does not return an $\epsilon$-best arm is at most $\delta/e$. Thus, if there are are at most $\frac{n^{3/8}}{4}$ arms that are $\alpha\epsilon$-close to $a^\star$ the algorithm terminates with an $\epsilon$-best arm with probability at least $1-\delta$.

Otherwise, there are at least $\frac{n^{3/8}}{4}$ arms that are $\alpha\epsilon$-close to $a^\star$. Since we select arms to $R$ u.a.r. and $|R| = \frac{n^{7/8}}{2}$ the likelihood of not having any arms that are $\alpha\epsilon$-close in $R$ is at most:

$$\left(1 - \frac{n^{3/8}}{4n}\right)^{|R|} = \left(1 - \frac{1}{4n^{5/8}}\right)^{\frac{n^{7/8}}{2}} = \left(\left(1 - \frac{1}{4n^{5/8}}\right)^{4n^{5/8}}\right)^{\frac{n^{1/4}}{8}} < e^{-\frac{n^{1/4}}{8}}$$

When we have $n > 10^5$ then $e^{-\frac{n^{1/4}}{8}} < \frac{1}{n^{1/4}}(1-1/e)$. Since $n > 1/\delta^4$ we get that the likelihood of an $\alpha\epsilon$-close to $a^\star$ not appearing in $R$ is smaller than $(1-1/e)\delta$. Let $\tilde{a}$ be an arm that is $\alpha\epsilon$-close to $a^\star$ in $R$. When we run NAÏVE ELIMINATION with approximation $(1-\alpha)\epsilon$ and $\delta/e$, we are guaranteed that with probability at least $1-\delta/e$ no arm that is $\epsilon$-far from $a^\star$ will have empirical mean higher than that of $\tilde{a}$. Since $\tilde{a}$ is $\alpha\epsilon$-close to $a^\star$ and $\alpha < 1$ this implies that the algorithm returns an arm that is at least $\epsilon$-close to $a^\star$ w.p. at least $1-\delta$ in this case as well. □

**Lemma 4.** *For any $\delta \leq 0.05$ ABA initialized with $\alpha = 1 - 1/e$ has sample complexity at most:*

$$\frac{18n}{\epsilon^2}\log\frac{1}{\delta}.$$

*Proof.* If $n < 1/\delta^4$ or $n < 10^5$ we invoke NAÏVE ELIMINATION and its sample complexity is $\frac{10n}{\epsilon^2}\log\frac{1}{\delta}$. To see this, notice that if $n < 1/\delta^4$ the sample complexity of NAÏVE ELIMINATION is:

$$\frac{2n}{\epsilon^2}\log\frac{n}{\delta} = \frac{2n}{\epsilon^2}\left(\log\frac{1}{\delta} + \log(n)\right) \leq \frac{2n}{\epsilon^2}\left(\log\frac{1}{\delta} + 4\log\frac{1}{\delta}\right) = \frac{10n}{\epsilon^2}\log\frac{1}{\delta}$$

If $n < 10^5$ then when $\delta < 0.05$ the sample complexity of NAÏVE ELIMINATION is:

$$\frac{2n}{\epsilon^2}\log\frac{n}{\delta} = \frac{2n}{\epsilon^2}\log\frac{1}{\delta}\left(1 + \frac{\log n}{\log\frac{1}{\delta}}\right) < \frac{10n}{\epsilon^2}\log\frac{1}{\delta}.$$

According to Claim 1 the sample complexity of AGGRESSIVE ELIMINATION with approximation $\epsilon$ and confidence $\delta$ is:

$$\left(1 + G(n,\delta)\right) \times \frac{2n}{\epsilon^2}\log\frac{1}{\delta}.$$

Therefore, when running with approximation $\alpha\epsilon$ and confidence $\delta' = \delta^{1+c}$ the sample complexity is:

$$\frac{1}{\alpha^2}\left(\frac{2n}{\epsilon^2}\log\frac{1}{\delta}\left((1+c)\left(1+G(n,\delta^{1+c})\right)\right)\right) \tag{22}$$

For any $\delta < 0.05$ we have that $\delta^{1+c} < \delta/2$ for $c = 1/4$. Thus, since we ran AGGRESSIVE ELIMINATION with confidence $\delta/2$ and $\alpha = 1 - 1/e$ the sample complexity is at most:

$$\frac{1}{\alpha^2}\left(\frac{2n}{\epsilon^2}\log\frac{1}{\delta}\left((1+c)\left(1+G(n,\delta^{1+c})\right)\right)\right) < \frac{1}{(1-1/e)^2}\left(\frac{2n}{\epsilon^2}\log\frac{1}{\delta}\left(\frac{5}{4}\left(1+G(n,\delta^{\frac{5}{4}})\right)\right)\right)$$

$$= \frac{10}{4(1-1/e)^2} \times \left(\frac{n}{\epsilon^2}\log\frac{1}{\delta}\left(\left(1+G(n,\delta^{\frac{5}{4}})\right)\right)\right)$$

$$< \frac{10}{4(1-1/e)^2} \times \left(\frac{n}{\epsilon^2}\log\frac{1}{\delta} \times 1.2\right)$$

$$< 8 \times \left(\frac{n}{\epsilon^2}\log\frac{1}{\delta}\right)$$

For the sample complexity of the NAÏVE ELIMINATION notice that it is applied on $A_T \cup R$ where $|A_T| = \frac{n^{3/4}}{2}$ and $|R| = \frac{n^{7/8}}{2}$. For any $n \geq 10^5$ we have that $\frac{n^{3/4}}{2} < \frac{n^{7/8}}{8}$ and therefore $|A_T \cup R| \leq \frac{5}{8} \cdot n^{7/8}$. Since $\alpha = 1 - 1/e$ the sample complexity of NAÏVE ELIMINATION is:

$$\frac{1}{(1-\alpha)^2}\left(\frac{5}{8}\frac{2n^{7/8}}{\epsilon^2}\log(\frac{5}{8}\cdot\frac{n^{7/8}}{\delta})\right) < \frac{1}{(1-\alpha)^2}\left(\frac{5}{8}\frac{2n^{7/8}}{\epsilon^2}\log\left(\frac{n^{7/8}}{\delta}\right)\right) \tag{23}$$

$$= \frac{10\cdot e^2}{8\cdot n^{1/8}}\left(\frac{n}{\epsilon^2}\log\frac{n^{7/8}}{\delta}\right) \tag{24}$$

$$= \frac{10\cdot e^2}{8\cdot n^{1/8}}\left(\frac{n}{\epsilon^2}\left(\frac{7}{2}\log(n^{-1/4})+\log\frac{1}{\delta}\right)\right) \tag{25}$$

$$< \frac{10\cdot e^2}{8\cdot n^{1/8}}\left(\frac{n}{\epsilon^2}\left(\frac{7}{2}\log\frac{1}{\delta}+\log\frac{1}{\delta}\right)\right) \tag{26}$$

$$= \frac{45\cdot e^2}{8\cdot n^{1/8}}\left(\frac{n}{\epsilon^2}\log\frac{1}{\delta}\right) \tag{27}$$

$$\leq \frac{45\cdot e^2}{8\cdot 10^{5/8}}\left(\frac{n}{\epsilon^2}\log\frac{1}{\delta}\right) \tag{28}$$

$$< 10\left(\frac{n}{\epsilon^2}\log\frac{1}{\delta}\right) \tag{29}$$

Therefore, the sample complexity of AGGRESSIVE ELIMINATION and NAÏVE ELIMINATION is smaller then:

$$18 \times \left(\frac{n}{\epsilon^2}\log\frac{1}{\delta}\right).$$

$\square$

## B.1 Asymptotic Sample Complexity

**Generalization of** $\phi(n)$**.** Recall that in our algorithm we condition on $n^{1/4} \geq 1/\delta$ and otherwise implement NAÏVE ELIMINATION$(A, \epsilon, \frac{\delta}{n})$. In general, $\forall d \geq 0$ if $n^d < 1/\delta$ we can $(\epsilon, \delta)$-learn the best arm using NAÏVE ELIMINATION$(A, \epsilon, \frac{\delta}{n})$ with sample complexity

$$\frac{2n}{\epsilon^2}\left(\log\frac{1}{\delta}+\frac{\log(n^d)}{d}\right) = 2\left(1+\frac{1}{d}\right)\frac{n}{\epsilon^2}\log\frac{1}{\delta}$$

For any choice of $d$ we can modify the AGGRESSIVE ELIMINATION to produce the same confidence and approximation guarantees under the assumption that $n^d \geq 1/\delta$, for any $d \in [0, \sqrt{n}]$. To do so all we need to do is make a modest modification in the definition of $\phi(n)$. Under an assumption $n^d \geq 1/\delta$ our definition of $\phi(n)$ was designed to satisfy the following inequality:

$$\exp\left(-\phi(n)^2\left(|A_i|-\frac{n^{3/8}}{4\phi(n)}\right)\right) \leq \frac{1}{10\cdot n^d\log n} \tag{30}$$

the left hand expression is the likelihood of the event that in an iteration $i$ the number of arms that are $\epsilon$-far from $a^\star$ that are overestimated is such that $a^\star$ is not included in the next round. The righthand expression becomes smaller than $\delta/(10\log(n))$ when $n^d > 1/\delta$.

We can therefore generalize the definition of $\phi(n)$ to $\phi(n,d)$ as follows:

$$\phi(n,d) = \sqrt{\frac{\log(10)+d\log(n)+\log\log(n)}{n^{3/4}}}$$

The larger $d$ is so is the sample complexity, but for $d = \sqrt{n}$ we get our desired asymptotic behavior. In particular get $\lim_{n\to\infty}\phi(n,d) = 0$, thus for any $\delta < 1$ we get $\lim_{n\to\infty}G(n,\delta) = 0$ and the number of rounds until the algorithm terminates $t(n) = \log n \times \left(\log\left(\frac{1}{\delta+\phi(n,d)}\right)^{-1}\right)$ approaches $\log n$ as well.

If $n^{\sqrt{n}} < \frac{1}{\delta}$ we may use the AGGRESSIVE ELIMINATION and getting a sample complexity of

$$\frac{2n}{\epsilon^2} \log \frac{n}{\delta} = \frac{2n}{\epsilon^2} \left( \log \frac{1}{\delta} + \log(n) \right) \leq \frac{2n}{\epsilon^2} \left( \log \frac{1}{\delta} + \log(\log^2(\frac{1}{\delta})) \right)$$

and there exist $\delta_0$ s.t. if $\delta < \delta_0$, the total sample complexity is $(2 + \lambda) \frac{n}{\epsilon^2} \log \frac{1}{\delta}$ for any $\lambda > 0$.

**Choosing $\alpha$ as a function of $n$.** The sample complexity of ABA is a convex combination of the sample complexity of AGGRESSIVE ELIMINATION and NAÏVE ELIMINATION:

$$\frac{1}{\alpha^2} \left( \frac{2n}{\epsilon^2} \log \frac{1}{\delta} \left( (1+c) \left( 1 + G(n, \delta^{1+c}) \right) \right) \right) + \frac{1}{(1-\alpha)^2} \left( \frac{5}{8} \frac{2n^{7/8}}{\epsilon^2} \log \left( \frac{5}{8} \cdot \frac{n^{7/8}}{\delta} \right) \right) \quad (31)$$

If we choose $\alpha = (1 - n^{-\frac{1}{16}})$ then as $n$ tends to infinity, in the limit the sample complexity is:

$$(1+c) \left( \frac{2n}{\epsilon^2} \log \frac{1}{\delta} \right)$$

Where we relied on the fact that for any fixed $\delta < 1$, $\lim_{n \to \infty} G(n, \delta^{1+c}) = 0$ for any choice of $c > 0$. Recall that we use $c$ to shrink $\delta$ so that instantiating AGGRESSIVE ELIMINATION with $\delta^{1+c}$ is guaranteed to include $a^\star$ in its output w.p. at least $(1 - 1/e)\delta$. As $\delta$ becomes smaller we require a smaller choice of $c$ as well. Thus, for any $c$ there exists a $\delta_0$ s.t. for any $\delta < \delta_0$ running AGGRESSIVE ELIMINATION with $\delta^{1+c}$ is guaranteed to include $a^\star$ in its output with probability at least $(1 - 1/e)\delta$.

**Theorem 6.** *For any $\lambda > 0$ there exist $\delta_0$ and $n_0$ s.t. for any $\delta < \delta_0$ and $n \geq n_0$, ABA $(\epsilon, \delta)$-learns a best arm with sample complexity at most:*

$$\left( 2 + \lambda \right) \frac{n}{\epsilon^2} \log \frac{1}{\delta}.$$

## C   Approximate Best Arm Likelihood Estimation by Hoeffding

**Lemma 3.** *Suppose $\lambda < 1$, $\delta \leq \delta_0$ where $\delta_0$ is the solution to $\frac{\lambda}{100} = \delta_0^{\lambda^2/256}$, and $n > 1/\delta$. If there are at most $n^{2/3}$ arms which are $\alpha\epsilon$-close to $a^\star$ then w.p. at least $1 - \frac{\delta}{2}$ we have that $a^\star$ is one of the $\frac{\lambda n}{50}$ highest estimated arms in $A$.*

*Proof.* First, we apply a Hoeffding bound on the estimation of $a^\star$. Suppose that we would like to estimate the value of $a^\star$ to accuracy $\epsilon \cdot (1 - \frac{\lambda}{16})\alpha$, with success probability at least $1 - \frac{\delta}{4}$. The number of samples this requires is

$$\frac{\log \frac{4}{\delta}}{2\alpha^2 \epsilon^2 (1 - \frac{\lambda}{16})^2} = \frac{\log \frac{4}{\delta}}{2\alpha^2 \epsilon^2 (1 - \frac{\lambda}{8} + \frac{\lambda^2}{256})} \leq \frac{\log \frac{1}{\delta}}{2\epsilon^2 \alpha^2 (1 - \frac{\lambda}{8})} \leq \frac{\log \frac{1}{\delta}}{2\epsilon^2 (1 - \frac{\lambda}{4})} = \left( 1 + \frac{\lambda}{2} \right) \frac{1}{2\epsilon^2} \log \frac{1}{\delta}$$

where the first inequality uses $\delta < \delta_0$ and the second one uses $\lambda < 1$. Since we have taken sufficiently many samples, the Hoeffding inequality applies.

For any other arm, we apply the Hoeffding bound to estimate its mean with accuracy $\epsilon \cdot \frac{\lambda}{16}$, but with failure probability $1 - \delta^{\lambda^2/256}$. The number of samples this requires is:

$$\frac{256}{2\epsilon^2 \lambda^2} \log \frac{1}{\delta^{\lambda^2/256}} = \frac{1}{2\epsilon^2} \log \frac{1}{\delta}$$

where we took the exponent out of the logarithm. Achieving this approximation and confidence is possible in this case as well since we are actually performing $(1 + \frac{\lambda}{2})\frac{1}{2\epsilon^2} \log \frac{1}{\delta}$ samples on each arm.

But since $\delta < \delta_0$, and $\frac{\lambda}{100} = \delta_0^{\lambda^2/256}$, this approximation is achievable when failure probability for each arm is bounded from above by $\frac{\lambda}{100}$. Hence the probability that we estimate more than $\frac{\lambda}{80}$ arms incorrectly is exponentially small in $n$. Since $n > 1/\delta$ this failure probability is at most $\frac{\delta}{4}$.

Taking a union bound over both events, we get that with probability at least $1 - \frac{\delta}{2}$ we have that $a^\star$ was estimated up to error $\epsilon(1 - \frac{\lambda}{8})\alpha$, and at most $\frac{\lambda n}{80}$ arms were estimated to error at least $\frac{\alpha\epsilon\lambda}{8}$. Condition on this event. Now there are two types of arms that we may estimate to be larger than $a^\star$:

- Arms which are $\epsilon\alpha$ close to $a^\star$: there are fewer than $n^{2/3} < \frac{3\lambda n}{400}$ such arms, since $n > \frac{1}{\delta_0}$;

- Arms which were estimated incorrectly: there are at most $\frac{\lambda n}{80}$ such arms.

As $\frac{\lambda n}{80} + \frac{3\lambda n}{400} = \frac{\lambda n}{50}$ and $|A_0| = \frac{\lambda n}{50}$, w.p. $\geq 1 - \frac{\delta}{2}$ the arm $a^\star$ is chosen to $A_0$. $\qquad\square$

**Lemma 5.** *For any $\lambda \in [0,1]$, $\delta \leq \delta_0$ where $\delta_0$ is the solution to $\frac{\lambda}{100} = \delta_0^{\lambda^2/64}$ suppose $n > 1/\delta$. Then* ABALEH *returns an $\epsilon$-best arm w.p. at least $1 - \delta$.*

*Proof.* Let $G$ denote the set of arms which are $\alpha\epsilon$ close to $a^\star$. We consider two cases. First, if $|G| < n^{2/3}$ then according to Lemma 3 with probability at least $1 - \frac{\delta}{2}$ we have $a^\star \in A_0$. Conditioning on this event, note that since $n > 1/\delta_0$ we have that $n^{2/3} < \left(\frac{\lambda n}{100}\right)^{3/4}$ and hence we can apply Claim 2 and deduce that with probability $1 - \frac{\delta}{4}$ we have that $A_T$ contains an $\epsilon\alpha$ approximate best arm. Finally, in this case with probability $1 - \frac{\delta}{4}$ we have NAÏVE ELIMINATION finds the an $(1-\alpha)\epsilon$ approximate best arm to an $\alpha\epsilon$ approximate best arm, which gives an $\epsilon$ best arm as required. Summing the errors and applying a union bound proves the lemma.

On the other hand, if $|G| \geq n^{2/3}$, then with probability $1 - 2^{-O(n^{1/6})} \geq 1 - \frac{\delta}{2}$ (since $n \geq 1/\delta$) we have that $T \cap G$ will be non empty. Again, with probability at least $1 - \frac{\delta}{4}$ NAÏVE ELIMINATION returns a $(1-\alpha)\epsilon$ approximate best arm to an $\alpha\epsilon$ approximate best arm, which gives an $\epsilon$ best arm as required. Again, a union bound shows that the probability of error is at most $\frac{\delta}{4} + \frac{\delta}{2} < \delta$. $\qquad\square$

**Sample complexity.** The sample complexity of ABALEH is the sum of the sample complexity of its three procedures:

1. The first iteration has sample complexity
$$\left(1 + \frac{\lambda}{2}\right)\left(\frac{n}{2\epsilon^2}\log\frac{1}{\delta}\right)$$

2. The sample complexity of calling AGGRESSIVE ELIMINATION$(A_0, \epsilon\alpha, \frac{\delta}{4})$ is
$$\frac{10\lambda n}{50 \cdot \epsilon^2 \alpha^2}\log\frac{4}{\delta} < \frac{99\lambda}{200}\left(\frac{n}{2\epsilon^2}\log\frac{1}{\delta}\right)$$
where we substituted $\alpha$ and used $\lambda < 1$;

3. Running NAÏVE ELIMINATION$(A_T \cup R, (1-\alpha)\epsilon, \frac{\delta}{4})$ when $n > 1/\delta_0$ has sample complexity at most:
$$\frac{2n^{3/4}}{\epsilon^2}\left(\log n + \log\frac{1}{\delta}\right) < \frac{\lambda}{100}\left(\frac{n}{2\epsilon^2}\log\frac{1}{\delta}\right)$$

## D  Lower Bound

**Theorem 7.** *For every $\beta > 0$ there exist $\epsilon_0, \delta_0$ such that for any algorithm in $\mathcal{F}$ which finds an $\epsilon$ best arm with success probability $1 - \delta$ where $\epsilon < \epsilon_0$, $\delta < \delta_0$, there exist $n_0$ such that if $n > n_0$, the algorithm requires at least $\left(\frac{1}{2} - \beta\right)\frac{n}{\epsilon^2}\log\frac{1}{\delta}$ queries.*

*Proof.* Suppose that there exists some algorithm $\mathcal{A} \in \mathcal{F}$ which uses less than $\left(\frac{1}{2} - \beta\right)\frac{n}{\epsilon^2}\log\frac{1}{\delta}$ queries. Then it must be that after $(1 + \nu) \times \frac{1}{\epsilon^2}\left(\frac{1}{2} - \beta\right)\log\frac{1}{\delta}$ iterations, $|S| \leq \frac{n}{1+\nu}$. But this means that $\mathcal{A}$ can succeed with the following task, with probability at least $1 - \delta$:

Given $m = (1 + \nu)\frac{1}{\epsilon^2}\left(\frac{1}{2} - \beta\right)\log\frac{1}{\delta}/$ samples on each arm, choose $\frac{1}{1+\nu}$ of the arms, such that this set contains an $\epsilon$ best arm. We will use $\nu = 0.0001\beta$.

Consider the following distribution: A bad arm is 0 w.p. $\frac{1}{2}$. and 1 $w.p.$ $\frac{1}{2}$. A good arm is 0 w.p. $p = \frac{1}{2} - \epsilon$ and 1 w.p. $1 - p$. There are $n - 1$ bad arms, and one good arm. Hence, $\mathcal{A}$ needs to identify $\frac{\nu}{1+\nu}$ of the arms, such that the good arm will not be in this set.

The optimal policy for this task given $(1 + \nu)m$ samples on each arm which maximizes the success probability, is to look at the number of zeroes each arm has, and to predict that the $\frac{\nu n}{1+\nu}$ arms which have the largest number of zeroes do not include the good arm. But the success probability of this policy can be bounded as follows:

For any $\xi > 0$ there exists $n_0$ such that if $n > n_0$ w.p. $1 - \xi$ there are at most $\frac{\nu n}{2}$ bad arms with more than $(1 + 0.001\beta\epsilon)\frac{m}{2}$ zeroes. We use $\xi = \frac{\delta}{2}$, which is easily satisfied by $n_0 = \frac{1000}{\beta^2 \epsilon^2 \delta^2}$.

Let $X_G$ be a random variable which denotes the number of zeroes of the good arm. We now bound the probability that the good arm will have too many zeros. That is, $\Pr[X_G > k]$ where $k = (1 + 0.001\beta\epsilon)\frac{m}{2}$. $X_G$ is the sum of random binomial variables, so we can apply a reverse tail bound to it.

According to [31], for $p \leq 1/2$ and $mp \leq k \leq m(1 - p)$ (which is indeed our case), it holds that

$$\Pr[X_G > k] \geq \Pr\left[Z > \frac{k - mp}{\sqrt{mp(1 - p)}}\right]$$

where Z is a normal $(0, 1)$ random variable.

We use a standard lower bound by [4] for upper tail of a normal random variables:

$$\Pr[Z > z] \geq \frac{z}{z^2 + 1} e^{-\frac{z^2}{2}}.$$

In our parameters, we have that $z = \frac{k - mp}{\sqrt{mp(1-p)}} = \frac{(1 + 0.0005\beta)m\epsilon}{\sqrt{m(1/4 - \epsilon^2)}} = \frac{2 + 0.001\beta}{1 - 4\epsilon^2}\epsilon\sqrt{m}$ which for $\epsilon < 0.0001\beta$ is more then $2\epsilon\sqrt{m} = 2\sqrt{(1 + \nu)\left(\frac{1}{2} - \beta\right)\log\frac{1}{\delta}}$. There exist $\delta_1$ such that for $\delta < \delta_1$ we have that $z$ is large enough for the following inequality to hold:

$$\frac{z}{z^2 + 1} e^{-\frac{z^2}{2}} \geq e^{-\frac{z^2}{2 - 0.001\beta}}.$$

Since $z = \frac{2 + 0.001\beta}{1 - 4\epsilon^2}\epsilon\sqrt{m} = \frac{2 + 0.001\beta}{1 - 4\epsilon^2}\sqrt{(1 + \nu)\left(\frac{1}{2} - \beta\right)\log\frac{1}{\delta}}$, then for $\epsilon < 0.0001\beta$, we have that $\frac{z^2}{2 - 0.001\beta} < (2 + 0.0001\beta)^2(1 + \nu)\left(\frac{1}{2} - \beta\right)\log\frac{1}{\delta}/(2 - 0.001\beta) < (1 - \beta)\log\frac{1}{\delta}$.

Combining the inequalities:

$$\Pr\left[X_G > (1 + 0.001\epsilon)\frac{m}{2}\right] \geq \delta^{1-\beta}.$$

However, if $\xi < \frac{\delta}{2}$ there exist $\delta_2$ such that if $\delta < \delta_2$, we have that

$$\delta^{1-\beta} - \xi > \delta.$$

This upper bounds the success probability of any algorithm in $\mathcal{F}$ making too few queries. Hence, for $\delta_0 < \min\{\delta_1, \delta_2\}$, $\epsilon_0 < 0.0001\beta$ and $n_0 = \frac{1000}{\beta^2 \epsilon^2 \delta^2}$ the theorem holds. $\square$

## E  Distributional Assumptions

Throughout the paper we use the assumption the the arms' Distributions are bounded in $[0, 1]$ in order to use the following version of the Hoeffding's inequality:

$$\Pr(\hat{X} - \mathbb{E}[X] \geq t) \leq e^{-2nt^2}$$

, where $n$ is the number of samples from a given arm, $X$ is the random variable for the sum of all of the samples from this arm and $\hat{X}$ is it its realization. The above bound holds for any sub-Gaussian distribution with a variance $\sigma^2$ which is smaller than some constant $\sigma_0^2$. Our results may be generalized for any sub-Gaussian distribution by scaling down the values and adjusting the selection of $\epsilon$, this will effect both the upper and lower bound in the same manner and the algorithmic results are still tight.

# F   Experiments

To illustrate the efficiency of the algorithms we conducted a simple numerical experiment. A reasonable concern may be that while our results suggest a dramatic improvement over the sample complexity of MEDIAN ELIMINATION this improvement may only be due to tighter analysis. In this section we rule out this possibility by experimentally comparing the actual sample complexity (not analysis) of our algorithms (SABA, ABA and ABALEH) with MEDIAN ELIMINATION and NAÏVE ELIMINATION. Note that all algorithms are guaranteed to $(\epsilon, \delta)$-learn the best arm, and thus our interest is in their sample complexity. Since our algorithms relative sample complexity improves as $n$ grows we were interested in observing this improvement emprically.

**Experimental setup.**   We fixed a choice of $\delta = 0.05$ and compared the sample complexity of all algorithms for $n = 300,000$ arms. Since all algorithms scale quadratically with $\epsilon$, we kept $\epsilon = 0.2$ in all our experiments[3]. The arms arms are distributed in the following way: $n - 1$ arms are Bernoulli random variables with mean $0.5$ and a single best arm is a Bernoulli random variable with mean $0.7 + 10^{-13}$.

**Results.**   We summarize the results in the table below

| Algorithm | Average number of samples for instance | Success (out of 1000 experiments) |
|:---:|:---:|:---:|
| MEDIAN ELIMINATION | $9.26 \cdot 10^9$ | 1000 |
| NAÏVE ELIMINATION | $2.34 \cdot 10^8$ | 1000 |
| SABA | $8.59 \cdot 10^6$ | 1000 |
| ABA | $1.98 \cdot 10^8$ | 1000 |
| ABALEH | $8.59 \cdot 10^7$ | 1000 |

SABA is making assumptions on the input (which hold for this scenario) and is 1000 times more efficient than MEDIAN ELIMINATION. Without assumptions on the input, ABALEH have a sample complexity which is 100 times more efficient than MEDIAN ELIMINATION. In fact, even the naive approach is more efficient than MEDIAN ELIMINATION.

# G   Technical overview

At a very high level, the idea behind the upper bounds is simple: for a given set of $n$ arms $A$, suppose we had an aggressive procedure to discard arms that terminates with a subset of arms $A_T \subset A$ s.t. $|A_T|$ is sublinear in $n$ and the optimal arm is in $A_T$, i.e. $a^\star \in A_T$. We could then run the most naive elimination procedure as discussed in the introduction to identify an approximate best arm [4] and the sample complexity would be small since $|A_T|$ is sublinear in $n$. Specifically, if the sample complexity of the aggressive elimination was $\frac{n}{2\epsilon^2} \log \frac{1}{\delta}$ the total sample complexity converges to $\frac{n}{2\epsilon^2} \log \frac{1}{\delta}$ as $n$ grows, since the naive elimination procedure is applied on sublinear number of arms.

The simple observation above is the philosophy behind the algorithms in this paper. The challenges are then how to design such aggressive elimination algorithms with low sample complexity that are guaranteed to not discard a best arm. Since we do not know how to actually design such algorithms, we instead design aggressive elimination algorithms with low sample complexity that are guaranteed to terminate with an arm that is sufficiently close the best arm. Since we are not guaranteed to have the best arm but a sufficiently close one, we then run the naive elimination procedure with higher precision and are guaranteed to return an $(\epsilon, \delta)$-best arm.

The main challenge that remains to discuss is how to produce an aggressive elimination procedure that is guaranteed to terminate with a set $A_T$ of sublinearly-many arms which includes an arm that is sufficiently close to the best arm $a^\star$. We do this by leveraging the fact that we have a large number of arms $n$. More specifically, the number of arms we discard in every step is a function of $n$. This then allows us to bound the likelihood of an approximately best arm being discarded in every iteration by a term that depends on $n$. Thus, as $n$ grows large, the likelihood that we fail to maintain an approximately best arm in $A_T$ vanishes.

To gradually introduce the ideas in this paper, we describe three algorithms: SIMPLE APPROXI-MATE BEST ARM (SABA), APPROXIMATE BEST ARM (ABA), and APPROXIMATE BEST ARM LIKELIHOOD ESTIMATION BY HOEFFDING (ABALEH). The presentation of SABA allows us to introduce the aggressive elimination procedure and the combination of this procedure with naive elimination. We then describe ABA which, unlike SABA, does not make assumptions about the input and allows the aggressive elimination procedure to terminate with an approximately best arm rather than a best one as in SABA. The last step is ABALEH which introduces a new technique for analyzing performance of best arm selection procedures. This technique then allows us to reduce the sample complexity of the aggressive elimination procedure from $\frac{2n}{\epsilon^2} \log \frac{1}{\delta}$ of ABA to the coveted $\frac{n}{2\epsilon^2} \log \frac{1}{\delta}$.

## H   Additional Related Work

The study of learning the best arm dates back to classic work by [8], and later by [2], [26], and [25]. More recently, $(\epsilon, \delta)$-PAC guarantees were studied in [11] and later by [12, 27]. There have since been other variants of this problem studied, including PAC learning a set of arms [5, 21, 24, 6], or the fixed budget setting where the goal is to minimize $\delta$ subject to a budget constraint on samples [5, 3, 13].

**Elimination Algorithms.**   A common approach for the $(\epsilon, \delta)$-PAC problem, is using algorithms who are based on elimination process such as the Median Elimination by [11] and [12]. In this framework, the algorithm may be described as series of rounds, where at each round we sample all non-eliminated arms and at the end of each round we may eliminate some of the arms until reaching a conclusion. Our work focuses on this family of algorithm and we show a lower bound for those algorithms that match our upper bound. Our lower bound hold for this class of algorithms.

**Lower bounds.**   [27] show that $\frac{n}{128\epsilon^2} \log \frac{1}{4\delta}$ samples are necessary for $(\epsilon, \delta)$-learning a best arm. As mention before, [9] show that their algorithm which is based on track-and-stop is tight instance-wise for arm distributions that comes from one-parameter one-dimensional canonical exponential families. The lower bound hold for any fixed number of arms as $\delta$ goes to 0. This lower bound is instance specific and it not clear on how to deduce worst case lower bound for all instances. Recently, [23] showed that $\Theta\left(\frac{n}{m}\right)$ samples are needed and sufficient when $n$ is the number of arms, $m$ is the number of $\epsilon$-best arms, and $\delta, \epsilon$ are constants.

**Implications**   Obtaining algorithms with dramatic lower sample complexity for a basic problem like learning a best arm can have several consequences. First, all previous algorithms that seek provable guarantees and directly employ MEDIAN ELIMINATION (e.g.  [20, 22, 32, 18, 7, 30]) can use the algorithms here instead and achieve dramatically lower sample complexity. From a practical perspective, MEDIAN ELIMINATION is not a particularly good choice. The naive sampling strategy of sampling each arm with approximation $\frac{\epsilon}{2}$ and confidence $\frac{\delta}{n}$ and selecting the arm with largest empirical mean $(\epsilon, \delta)$-learns a best arm and has lower sample complexity than MEDIAN ELIMINATION whenever the number of arms is smaller than $2^{1500}$. Nevertheless there is a great deal of work on heuristics based on MEDIAN ELIMINATION. Our hope is that some of the ideas presented here would not only contribute to *provably* learning a best arm, but also heuristics.

## I   Running Time and Implementation

The algorithm by [9] reaches the optimal sample complexity for many interesting distributions but it's main drawback is in aspects of computation time. The Implementation of their algorithm requires to solve many min-max problems iteratively. Hence having a computation time which is at least squared in the number of arms.

A main advantage of elimination algorithms compared to other approaches is the possibility of sampling all (non-eliminated) arms in parallel. In other words, having low adaptive complexity. [19, 1] show that $(\epsilon, \delta)$-PAC can be solved using $\Theta(\log^*_{1/\delta} n)$ rounds. The number of rounds in the algorithms that we present is $\Theta(\log^*_{1/\delta} n)$, which is still small compared to algorithms which are not elimination algorithms and are usually full adaptive.

All of the algorithms that we present are elimination algorithm which are based on following process. At each round a subset of the $n$ arms is sampled where each armed is sampled at most $\frac{2}{\epsilon^2} \log \frac{n}{\delta}$

times. After the sampling process, we keep as possible candidates, a fraction of the arms with the highest mean. The number of samples at each round is a function the round number, and the fraction of remaining arms is a function of the arms at the beginning of this round. Both of which can be computed efficiently. The filtering process can be computed in $O(\log^2 n)$ time in PRAM model [16], hence giving a total implementation in poly-logarithmic time complexity which is an exponential improvement compared to [9].