[Reviews · NeurIPS 2020]

Review 1

Summary and Contributions: The paper studies the best arm identification problem: given a set of n arms, each associated with a distribution supported on [0, 1], the goal is to identify an arm with the highest mean. More precisely, given positive epsilon and delta, the goal is to identify any arm whose mean is within epsilon of the highest mean, with success probability at least 1 - delta. How many arm pulls are needed to achieve this? This paper gives two algorithms for this task, for different ranges of n and delta. They also show that, for fixed epsilon and delta, one of their algorithms has asymptotically optimal sample complexity among the class of elimination-based algorithms.

Strengths: This is a fundamental problem. The proposed algorithms are simple and elegant and improve over the known algorithms by constant factors, as shown by theoretical and experimental results. One of the proposed algorithms is asymptotically optimal (even its constant factor).

Weaknesses: The lower bound holds for elimination-based algorithms only. There is no unified algorithm covering all ranges of n, epsilon, and delta and achieving the optimal sample complexity.

Correctness: I haven’t verified the proof details but the ideas seem robust.

Clarity: Yes, I was able to follow the paper easily. It’s a pleasure to read and gives intuition about the proof ideas.

Relation to Prior Work: Yes, even though I was not familiar with the prior work on this problem before reading this paper.

Reproducibility: Yes

Additional Feedback: Thanks for the rebuttal. I have kept my score unchanged. First, the supplementary material hasn’t complied correctly; please fix the question marks. The results in main contribution section are not explained clearly, in particular about the relationship between n, delta, epsilon, and what is approaching zero/infinity. In result 1, is epsilon or delta fixed? Is n approaching infinity or both n and 1/delta are approaching infinity? Please first state your result clearly and only then compare it with previous work. In contribution 2, what is meant by “n is independent of delta”? Do you mean one of them is fixed and the other one approaches infinity? Or it holds for all values of n and delta? Is epsilon and delta fixed, and n approaches infinity? What do you mean by “converge”? What’s the limiting variable? For contribution 3, “arbitrary close” is not clear. I would suggest writing the exact mathematical statement, as it is the most clear way to express this result. Also, the section title must be main contributions as there are more than one contribution. In line 206, “will include … w.p. at least” must be “will not include … with probability at most”. Please do not use the nonstandard abbreviation w.p. without defining it. Since you’re referring to Hoeffding’s bound several times, it’s worth writing it down explicitly in the main part of the paper. For given values of n, epsilon, and delta, what is the best algorithm to use according to your best knowledge and among the algorithms you introduce in this paper? A simple answer to this question will be much appreciated by a practitioner who does not have the time/knowledge to read and understand your theorems.


Review 2

Summary and Contributions: The paper considers the multi-armed bandit problem of identifying an arm within epsilon of the largest mean with probability at least 1- delta. The authors focus on the regime where the number of arms n is larger than the failure probability delta. Although median elimination is optimal up to a constant, this constant is quite large and the paper provides an algorithm with a much better constant, improving by a factor of 1500.

Strengths: The problem is well motivated. It is a fundamental theoretical problem, and the poor constants of median elimination do beg the question. Furthermore, because the algorithm can be applied to many problems as a subroutine (e.g., best arm identification), it could prove to be quite useful. The approach and analysis are novel and interesting. I think that the experiments in the supplementary material are compelling because they show the improvement over medium elimination. 

Weaknesses: I think the paper could make a stronger case for its practical applications. It would be interesting to see if using this algorithm is a subroutine improves the performance of some best arm identification algorithms (e.g., exponential gap elimination). This would be useful for showing the benefits of this algorithm for applications. Their strongest results hold for as n grows to infinity. On the other hand, Theorem 1 seems to give their guarantee for any value of n. Here, the constant is 18, which is much larger than their lower bound. Thus, it seems to remain an open question what is the optimal sample complexity for a particular value of n and delta.

Correctness: Yes.

Clarity: It is a well-written paper.

Relation to Prior Work: Yes.

Reproducibility: Yes

Additional Feedback: It would be useful if the authors summarized succintly the central insights that led to these results. After rebuttal: I have read the response and other reviews and my score remains the same.


Review 3

Summary and Contributions: This paper considers the problem of identifying an epsilon good arm with high confidence. They propose an algorithm whose sample complexity converges to n/2epsilon^2log(1/delta) asymptotically (under an assumption that \delta>1/n), and demonstrate a similar lower bound. They also provide improved results for the setting when n is independent of delta.

Strengths: The paper addresses an important problem of finding an epsilon good arm and provides a tight result over many regimes of n and delta for elimination algorithms. As far as recommendation for publication. The results are theoretical however, I tend towards acceptance due to the completeness of the results in this paper.

Weaknesses: I feel that though the analysis is deep and technical the result may not be immediately applicable. The range of \delta and n proposed are not practical for many settings. If the authors really feel that this algorithm improves over naive elimination (which suffers a bad union bound over n), then they should demonstrate this empirically - in general experiments would have helped the paper. One concern that I had is that the paper seemed focused on nailing tight asymptotic factors for elimination algorithms. However, there are tight instance dependent lower bounds (for example ttp://www.jmlr.org/papers/volume17/kaufman16a/kaufman16a.pdf remark 5). If many arms are much further than epsilon away, then this bound can be much tigher. Similarly, if many arms are epsilon good, we should not need a large number of samples - the goal of [23] in the unverifiable regime. It feels like a more useful line of work is improving these instance specific bounds.

Correctness: I did not check the proofs carefully but it seems correct.

Clarity: Yes.

Relation to Prior Work: Yes. The idea of removing a constant fraction everytime is

Reproducibility: Yes

Additional Feedback: ##### After Feedback ##### I am happy with the authors feedback and it addresses my concerns. I'll update my review accordingly.


Review 4

Summary and Contributions: This paper characterizes the worst-case sample complexity of identifying an epsilon-best arm with specified confidence with the correct constant factors. The authors give an elimination-style algorithm that has much tighter constant factors than the order-optimal median elimination algorithm, and show via simulation on a small problem instance that their algorithm indeed requires fewer samples than median elimination and other baselines. With a lower bound, they show the sample complexity is asymptotically optimal as the number of arms grows.

Strengths: It is known that median elimination has large constant factors in its sample complexity, authors have managed to provide better algorithms. The results are significant, novel and relevant to the community.

Weaknesses: While not necessary, an empirical evaluation of the type in Appendix H could be included in the main text. Empirically, it would be useful to also show the performance of a track-and-stop algorithm (Ref [14]) and a upper confidence bound algorithm (Ref [17]).

Correctness: I have checked some of the claims in the appendix and they look correct to me.

Clarity: The paper is very well-written and easy to follow. Some minor typos: *line 3 of algorithm 2: shouldn't it say "\lfloor |A_i| (delta + phi(n)) \rfloor" *third argument of NaiveElimination: should it be "delta" in line 181

Relation to Prior Work: There is adequate discussion in the related work.

Reproducibility: Yes

Additional Feedback: After reading author response: I have read the other reviews as well, and I don't need to change my score.

[Author Response · NeurIPS 2020]

We thank all the reviewers for their insightful comments and suggestions.

**Reviewer 1.** Thank you very much for your review and helpful suggestions. Detailed response is included below.

• "The results in main contribution section are not explained clearly, in particular about the relationship between $n$, $\delta$,
$\epsilon$, and what is approaching zero/infinity.": Thank you for this comment. We'll add to each bullet point in the main
contributions in the introduction a formal statement as in theorems 1-4 so that the relationships between the problem
parameters are formally stated already at the introduction.
• "For given values of $n, \epsilon$, and $\delta$, what is the best algorithm to use according to your best knowledge...?": Generally
speaking, ABALEH has best sample complexity whenever $n > 10^5$ and $n > 1/\delta$. When these conditions do not hold,
the naive approach should be taken. SABA (which make assumptions on the input) and ABA are used mostly for
didactic purposes to present and analyze the construction of ABALEH. We will discuss this in the body of the paper.
• Regarding all other comments: supplementary material compilation, line 206 and Hoeffding's bound. Thank you
very much. We'll fix compilation and add the Hoeffding bound in the main body of the paper.

**Reviewer 2.** Thank you very much for your review and helpful suggestions. Detailed response is included below.

• "It would be interesting to see if using this algorithm is a subroutine improves the performance...": Agreed. That's
an excellent idea and we would add such an analysis for gap elimination and other algorithms using MEDIAN
ELIMINATION as a subroutine. Indeed, the complexity of such algorithms largely depends on MEDIAN ELIMINATION,
thus as the results in appendix H. our algorithms will make a substantial improvements in these settings as well.
• "It would be useful if the authors summarized succinctly the central insights that led to these results": This is a great
idea, and we will add such a summary as a technical overview in the introduction.

**Reviewer 3.** Thank you very much for your review and helpful suggestions. The comments seem to stem from parts of
the paper that were overlooked. Empirical evaluation was indeed performed and can be found in Appendix H and we
included a discussion about instance-based analysis and why it is not applicable for the PAC setting studied here. We
elaborate further in the comments below and hope you will consider revising your score based on this response.

• "though the analysis is deep and technical the result may not be immediately applicable. The range of $\delta$ and $n$
proposed are not practical for many settings. If the authors really feel that this algorithm improves over naive
elimination (which suffers a bad union bound over n), then they should demonstrate this empirically - in general
experiments would have helped the paper.": Perhaps it has been overlooked, but **Appendix H is dedicated to**
**empirical evaluation**. It shows dramatic benefit of ABALEH, even for reasonable values of $n$ and $\delta$.
• Regarding concerns when comparing to instance dependent results: **Please see paragraph starting on line 88 titled:**
**"'From Instance-based to worst case analysis"**. In particular, one of the challenges with comparing PAC bounds
to instance specific bounds, is that instance specific algorithms assume that $n$ is **constant** and $\delta$ goes to zero, but do
not have a simple closed form expression which is based only on $n$ that determines the rate at which $\delta$ must go to zero
to make the analysis work, and what happens at given (finite) values of $n$ and $\delta$. As a concrete example for how this
is a problem, if $\delta \ll 1/n$, then we need to worry about the $\log 1/\delta$ more than about the $\log n$ in naive elimination.
OTOH, if $\delta \ll 2^n$, we need to worry about the $\log(1/\delta)$ factor more than about the $n$ factor in naive elimination
- this is the $n$ factor that instance optimal algorithms try to save in the first place (the biggest difference between
instance optimal algorithms and worst case algorithms like ours is when there is a unique best arm, one arm which is
almost $\epsilon$-close, and $n - 2$ arms which are always zero). Moreover, the gain in instance-based algorithms is bounded
not just by $n$ (which is assumed to be constant) but also by $1/\epsilon^2$, since their gain comes from the difference between
one $\epsilon$-far arm (which makes it difficult for the worst case algorithm) and the other arms that can be always zero, and
the smaller epsilon is the largest this difference is. To summarize, both research directions on instance-based and
worst case (i.e. PAC) learning algorithms are valid, but are useful for completely different parameter domains.

**Reviewer 4.** Thank you for your time and efforts.

• "...an empirical evaluation... could be included in the main text. Empirically, it would be useful to also show the
performance of a track-and-stop algorithm (Ref [14]) and a upper confidence bound algorithm (Ref [17]).": Thank
you for this comment. We will revise the manuscript to include the empirical evaluation of appendix H in the main
body of the paper. Regarding the track-and-stop algorithm, we ran code provided us by authors of these papers.
Unfortunately, the track-and-stop algorithms can only be run for small values of $n$ (roughly $n = 100$). We discuss
the inherent implementation and running time bottlenecks of track-and-stop in appendix G.
• "line 3 of algorithm 2: shouldn't it say "$\lfloor |A_i|(\delta + \phi(n)) \rfloor$"": The line as written is technically correct but writing it
as you suggest is clearer and we will change it – thank you.
• "third argument of NaiveElimination: should it be "$\delta$" in line 181": True. Thank you.

[Meta-Review · NeurIPS 2020]

This paper studies the PAC best-arm identification problem where an algorithm seeks to return an arm with a mean within epsilon of the best with probability at least 1-delta using as few total pulls as possible. It was known almost a decade ago that this was possibly with O(n epsilon^{-2} log(1/delta) ) pulls using the celebrated median elimination algorithm. However, median elimination has a constant in the thousands and is not practical. This paper seeks the optimal constant as n -> infinity. While finite-sample results are proven, the asymptotic regime is of interest because it is shown in the minimax regime that the leading constant in their upperbound is tight. Authors raised concerns that this instance-independent sample complexity can be much larger than instance-dependent sample complexities, and there exist algorithms that achieve optimal instance-dependent sample complexities, at least asymptotically. However, the novelty of the solution and fundamental importance of the problem merit acceptance.